# Nifuroxazide suppresses PD-L1 expression and enhances the efficacy of radiotherapy in hepatocellular carcinoma

**Tiesuo Zhao[1,2,3†], Pengkun Wei[1,2,4†], Congli Zhang[1,2], Shijie Zhou[1,2], Lirui Liang[1,2], Shuoshuo Guo[1,2], Zhinan Yin[5], Sichang Cheng[1,2], Zerui Gan[1,2], Yuanling Xia[1,2], Yongxi Zhang[6], Sheng Guo[1,2], Jiateng Zhong[2], Zishan Yang[1,2], Fei Tu[2], Qianqing Wang[7,8], Jin Bai[7,8], Feng Ren[3]\*, Zhiwei Feng[1,2]\*, Huijie Jia[2]\***

[1]Department of Immunology, School of Basic Medical Sciences, Xinxiang Medical University, Xinxiang, China; [2]Xinxiang Engineering Technology Research Center of immune checkpoint drug for Liver-Intestinal Tumors, Xinxiang Medical University, Xinxiang, China; [3]Henan International Joint Laboratory of Immunity and Targeted Therapy for Liver-Intestinal Tumors, Xinxiang Medical University, Xinxiang, China; [4]Zhengzhou Central Hospital Affiliated to Zhengzhou University, Zhengzhou, China; [5]The Biomedical Translational Research Institute, Faculty of Medical Science, Jinan University, Guangzhou, China; [6]Department of Oncology, The Third Affiliated Hospital of Xinxiang Medical University, Xinxiang, China; [7]Department of Gynecology, Xinxiang Central Hospital, Xinxiang, China; [8]The Fourth Clinical College, Xinxiang Medical University, Xinxiang, China

**\*For correspondence:**
renfeng@xxmu.edu.cn (FR);
123066@xxmu.edu.cn (ZF);
zhongziqi1115@163.com (HJ)

[†]These authors contributed equally to this work

**Abstract** Radiation therapy is a primary treatment for hepatocellular carcinoma (HCC), but its effectiveness can be diminished by various factors. The over-expression of PD-L1 has been identified as a critical reason for radiotherapy resistance. Previous studies have demonstrated that nifuroxazide exerts antitumor activity by damaging the Stat3 pathway, but its efficacy against PD-L1 has remained unclear. In this study, we investigated whether nifuroxazide could enhance the efficacy of radiotherapy in HCC by reducing PD-L1 expression. Our results showed that nifuroxazide significantly increased the sensitivity of tumor cells to radiation therapy by inhibiting cell proliferation and migration while increasing apoptosis in vitro. Additionally, nifuroxazide attenuated the up-regulation of PD-L1 expression induced by irradiation, which may be associated with increased degradation of PD-L1 through the ubiquitination-proteasome pathway. Furthermore, nifuroxazide greatly enhanced the efficacy of radiation therapy in H22-bearing mice by inhibiting tumor growth, improving survival, boosting the activation of T lymphocytes, and decelerating the ratios of Treg cells in spleens. Importantly, nifuroxazide limited the increased expression of PD-L1 in tumor tissues induced by radiation therapy. This study confirms, for the first time, that nifuroxazide can augment PD-L1 degradation to improve the efficacy of radiation therapy in HCC-bearing mice.

## eLife assessment

This **valuable** study evaluates the effects of nifuroxazide on radiotherapy for the treatment of hepatocellular carcinoma. **Solid** evidence is provided to support the conclusion that nifuroxazide facilitates the downregulation of PD-L1 and may improve therapy outcomes when combined with radiotherapy, though the inclusion of additional cell lines and animal models would have strengthened the study. This work will be of interest to cancer biologists and those working in immuno-oncology.

## Introduction

Cancer continues to be the primary cause of death and a major public health issue globally (*Siegel et al., 2021*). Among the various types of cancer, HCC has exhibited an increasing incidence and morbidity, and it is estimated that the number of new HCC cases will surpass one million by 2025 (*Feng et al., 2019*; *Anonymous, 2021*). The current therapeutic options for HCC comprise surgery, targeted therapy, radiation therapy, chemotherapy, and immunotherapy, among others. Nevertheless, the majority of patients still experience an unfavorable prognosis. Consequently, the development of effective treatment strategies for HCC is a pressing and challenging issue (*Tomášek and Kiss, 2020*).

Radiation therapy is a widely used localized therapeutic approach for the treatment of HCC (*Klein and Dawson, 2013*; *Chen et al., 2021*). It has been demonstrated that radiation therapy can induce DNA damage and mutations in tumor cells through the generation of X-rays or γ-rays, leading to the death of malignant cells (*Barber, 2015*). Despite significant progress in recent decades, the effectiveness of radiation therapy is limited by various factors. Studies have shown that radiation therapy can contribute to the formation of an immunosuppressive tumor microenvironment (TME) by promoting the infiltration of M2 macrophages, increasing the number of regulatory T cells, impairing the function of CD8 + T lymphocytes, and inducing the release of immunosuppressive cytokines (*Shevtsov et al., 2019*). Therefore, improving the immunosuppressive state of HCC may enhance the clinical efficacy of radiation therapy.

Programmed cell death 1 ligand 1 (PD-L1) is a transmembrane glycoprotein predominantly expressed in T lymphocytes, B lymphocytes, macrophages, and tumor cells. Upon binding with programmed cell death protein 1 (PD-1), PD-L1 generates inhibitory signals that suppress the proliferation of T lymphocytes and accumulation of antigen-specific T cells in lymph nodes (*Sato et al., 2020*; *Voli et al., 2020*; *Wang et al., 2020*). Radiation therapy induces DNA damage and repair, activating and up-regulating PD-L1, and creating an immune-suppressive microenvironment, which is a major contributor to the radiotherapy resistance of tumors (*Azad et al., 2017*; *Gong et al., 2017*). Furthermore, tumor-associated macrophages (TAMs) also take part in shaping the tumor microenvironment by expressing PD-L1. In mice with tumors, inhibition of the PD-1/PD-L1 pathway significantly stimulated the activation of macrophages (*Hartley et al., 2018*; *Sumitomo et al., 2019*). Similarly, in HCC, radiation therapy up-regulates the expression of PD-L1 via the IFN-γ-Stat3 signaling pathway (*Greten and Sangro, 2017*). Therefore, blocking the PD-1/PD-L1 pathway may be an effective systemic treatment for enhancing the efficacy of radiation therapy.

The PD-1/PD-L1 pathway blockade has been shown to provide satisfactory clinical benefits for most tumor types (*Salmaninejad et al., 2019*). However, the development and research of new drugs are time-consuming, costly, and uncertain (*Scannell et al., 2012*), and exploring the potential indications of old drugs has been endorsed by scholars (*Wu, 2013*). Nifuroxazide, an antidiarrheal drug, has been demonstrated to significantly inhibit the activation of Signal Transducer and Activator of Transcription 3 (Stat3) and has also exhibited effects in promoting the death of multiple myeloma cells and inhibiting tumor growth. Importantly, treatment with nifuroxazide has shown low toxicity that does not damage peripheral blood mononuclear cells (*Nelson et al., 2008*). However, the relationship between nifuroxazide and the expression of PD-L1 remains unknown, as well as whether nifuroxazide can enhance the efficacy of radiation therapy in tumors by blocking the expression of PD-L1.

We have revealed that nifuroxazide exerts a significant effect on promoting the degradation of PD-L1 in HCC cells. Notably, administration of nifuroxazide led to a reduction in the expression of PD-L1 in mice, which in turn enhanced the sensitivity of radiation therapy in mice with HCC. This approach may offer a promising combination strategy to counteract the radio-resistance of HCC.

## Results

### The administration of Nifuroxazide augmented the radiosensitivity in the treatment of HCC

The study initially evaluated the impact of Nifuroxazide on HCC radiosensitivity by assessing cell proliferation, migration, and apoptosis. The CCK-8 assays showed that cell proliferation was significantly inhibited after adding Nifuroxazide for 24 hr, and the inhibitory effect was further enhanced when Nifuroxazide was added after radiotherapy (*Figure 1A* and *Figure 1—figure supplement 1A*). Interestingly, after 48 hr of radiotherapy, there was no significant inhibition of cell proliferation, but

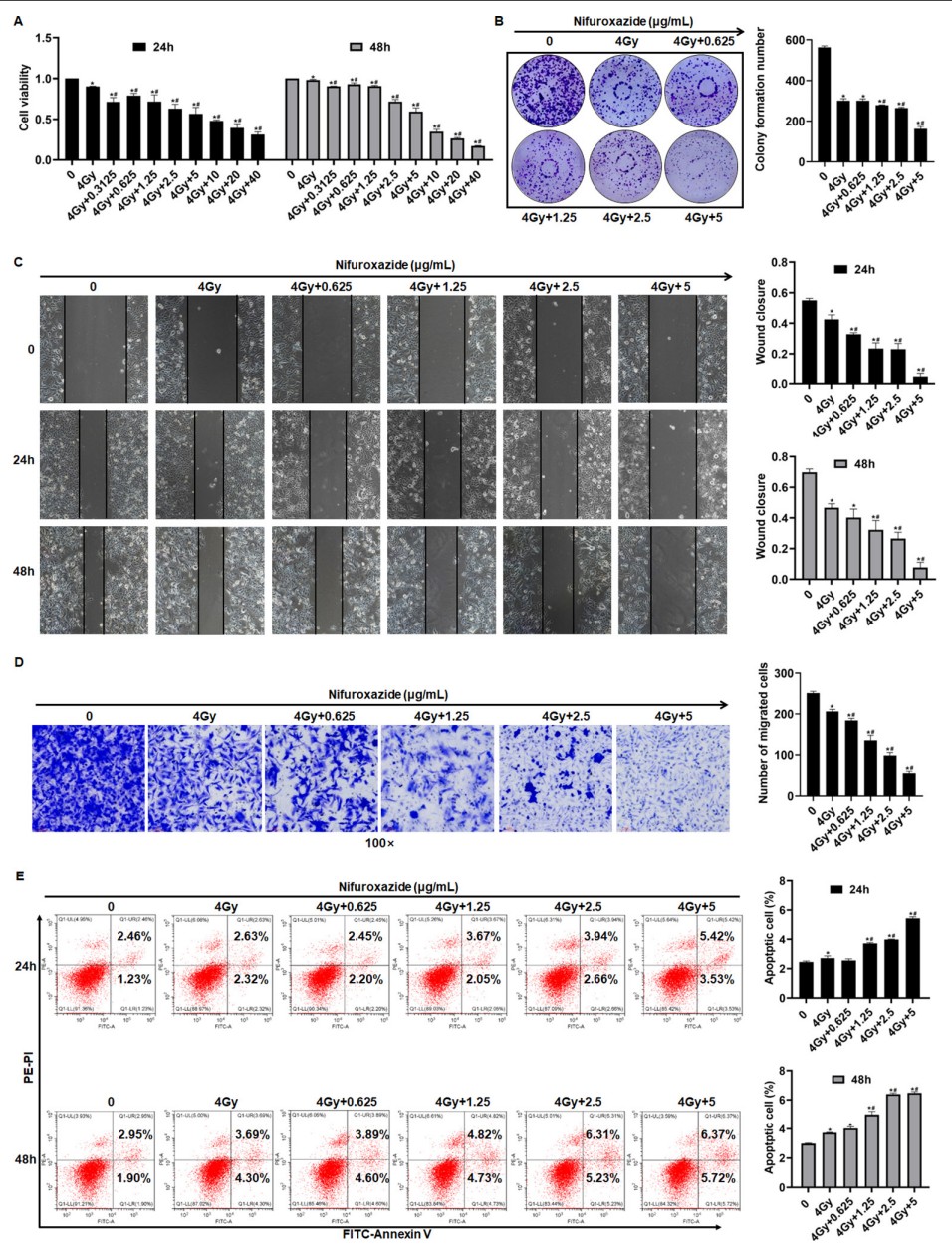

**Figure 1.** The effect of radiotherapy in combination with nifuroxazide on the proliferation, migration, and apoptosis of HepG2 cells. (**A**) The effect of the radiotherapy in combination with nifuroxazide on the viability of HepG2 cells by CCK-8 assay. (**B**) The effect of the radiotherapy in combination with nifuroxazide on the proliferation of HepG2 cells by cell clone formation assay. (**C**) The effect of the radiotherapy in combination with nifuroxazide on the migration of HepG2 cells by Wound-Healing assay. (**D**) The effect of the radiotherapy in combination with nifuroxazide on the migration of HepG2 cells by transwell assay. (**E**) The effect of the radiotherapy in combination with nifuroxazide on the apoptosis of HepG2 cells by flow cytometry assay. One-way analysis of variance (ANOVA) was carried out and the data were presented as mean± SD (n=3). Compared with the control group, *p<0.05; compared with '4 Gy' group, #p<0.05.

The online version of this article includes the following figure supplement(s) for figure 1:

**Figure supplement 1.** The effect of radiotherapy and nifuroxazide on the proliferation, migration, and apoptosis of HepG2 cells.

co-treatment with Nifuroxazide at concentrations of 2.5, 5, 10, 20, and 40 μg/ml resulted in evident inhibition. The combination of Nifuroxazide and radiotherapy also significantly inhibited cell proliferation, as demonstrated by cell cloning results (*Figure 1B*). Additionally, the wound-healing assay and transwell assay showed substantial hindrance of cell migration at both 24 and 48 hr after radiotherapy. The wound healing assay demonstrated that in comparison with the control group, the migratory ability of cells was significantly reduced after treatment for 24 hr in the Nifuroxazide alone group, the radiotherapy alone group, and the combination group (Nifuroxazide plus radiotherapy, showing the most significant reduction) (*Figure 1C* and *Figure 1—figure supplement 1B*). Similarly, the inhibition of cell invasion was further enhanced when the cells were co-treated with Nifuroxazide and radiotherapy (*Figure 1D*). The combined treatment also resulted in a more potent pro-apoptotic effect on the cells. *Figure 1E* and *Figure 1—figure supplement 1C* indicated that radiotherapy alone did not have a pro-apoptotic effect on HCC, but the combination of radiotherapy and Nifuroxazide at concentrations of 1.25, 2.5, and 5 μg/ml exhibited significant pro-apoptotic effects. These findings suggest that Nifuroxazide may serve as a promising candidate drug to improve the radiosensitivity of HCC treatment.

## Nifuroxazide had a significant effect on the expression of proteins associated with cell proliferation and apoptosis

To investigate the underlying mechanism of Nifuroxazide's ability to enhance HCC radiosensitivity, the study analyzed the expression of proteins associated with proliferation and apoptosis in HCC cells. Activation of the oncogene Stat3, which plays a critical role in tumor development, was effectively inhibited by radiotherapy in cells. Moreover, Nifuroxazide, an inhibitor of Stat3, was able to suppress the activation of Stat3 in HepG2 cells that had undergone radiotherapy, as demonstrated in *Figure 2A*. MMP2 plays an important role in cancer cell migration. The results showed that the combination therapy showed a significant inhibiting effect on the expression of MMP2 (*Figure 2B*). The expressions of PCNA and Ki67, which are closely linked to cell proliferation, were reduced by radiotherapy. Furthermore, when administered in conjunction with Nifuroxazide, this inhibitory effect was even more pronounced, as depicted in *Figure 2C and D*. The combination therapy also suppressed the expression of cyclin D1, a protein involved in the cell cycle, as shown in *Figure 2E*. In addition, the study examined apoptotic proteins, as illustrated in *Figure 2F–N*. Caspase 3 is the primary enzyme responsible for the cleavage of cells during apoptosis. The results showed that the expression of cleaved-caspase 3 (c-caspase 3) was significantly increased at 24 or 48 hr after radiotherapy. Furthermore, the combination of radiotherapy and nifuroxazide resulted in a more pronounced upregulation of c-caspase 3, indicating a significant increase in cell apoptosis in HCC. Apoptosis can be induced through various pathways, with the mitochondrial apoptosis pathway being the most important. The results showed that the combination of radiotherapy and nifuroxazide had a significant impact on the expression of Bax, Bcl-2, and cytochrome C. Specifically, the combination therapy inhibited the expression of Bcl-2, which led to the translocation of Bax to mitochondria and the release of cytochrome C into the cytoplasm. The release of cytochrome C subsequently activated caspase 9, another pro-apoptotic protein. These events indicate that radiotherapy in combination with nifuroxazide significantly enhances apoptosis in HCC. PARP is the substrate of caspases and plays a vital role in regulating apoptosis. The activation of PARP is considered an important indicator of cell apoptosis and caspase 3 activation. The results demonstrated that the combination therapy increased the expression of c-PARP, suggesting that it had significant capabilities in inducing apoptosis. Overall, these findings suggest that the combination of radiotherapy and nifuroxazide can inhibit proliferation and increase apoptosis in HCC by regulating different tumor-related proteins.

## Radiotherapy in combination with nifuroxazide significantly inhibited the growth of tumors in tumor-bearing mice and prolonged their survival period

A tumor-bearing mice model was used to validate the anti-tumor effect of the combination treatment. The treatment protocol for the study is illustrated in *Figure 3A*. Seven days after the final treatment, the tumors were extracted, and the size and weight of the tumors were measured. The results showed that both the radiation therapy and nifuroxazide treatment groups exhibited significant inhibition of tumor growth compared to the PBS group, as evidenced by smaller tumor size and weight.

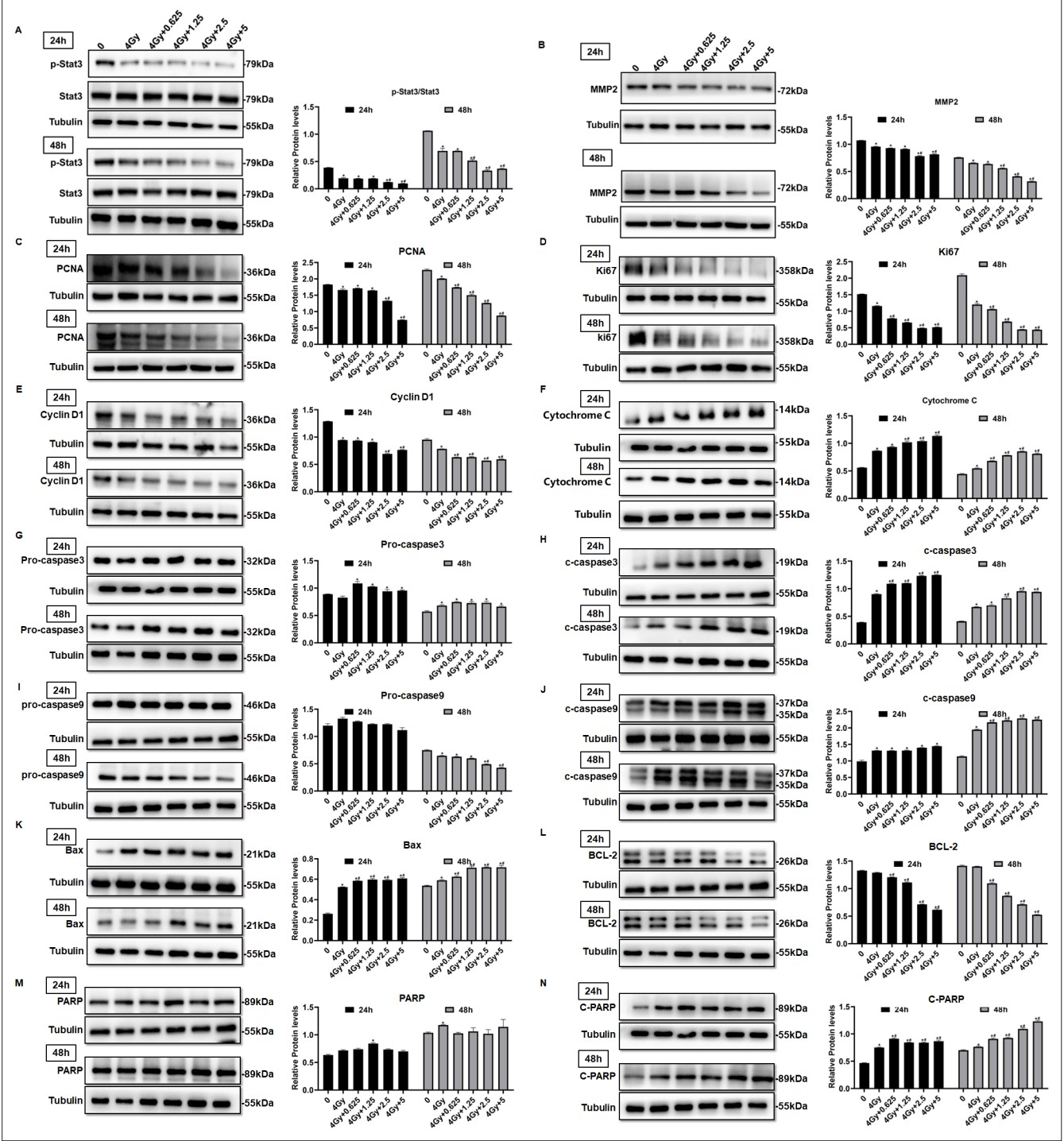

**Figure 2.** The effect of radiotherapy in combination with nifuroxazide on the expressions of tumor-associated proteins in cells. After radiotherapy, HepG2 cells were treated with nifuroxazide at the different dose. At 24 hr or 48 hr after being incubated, the expression of tumor-associated proteins was detected by Western blot. (**A–E**) The expression of Stat3, p-Stat3, PCNA, Ki67, and cyclin D1 related to cell proliferation was analyzed by Western blot. (**F–N**) The expression of cytochrome C, pro-caspase 3, c-caspase 3, pro-caspase 9, c-caspase 9, Bax, Bcl-2, PARP, and c-PARP related with cell apoptosis was analyzed by Western blot. One-way analysis of variance (ANOVA) was carried out and the data are presented as mean ± SD (n=3). Compared with the control group, *p<0.05; compared with '4 Gy' group, #p<0.05.

The online version of this article includes the following source data for figure 2:

**Source data 1.** Original file for the Western blot analysis in *Figure 2A-F* (anti-p-Stat3, anti-Stat3, anti-MMP2, anti-PCNA, anti-Ki67, anti-Cyclin D1, anti-Cytochrome C, and anti-tubulin).

*Figure 2 continued on next page*

*Figure 2 continued*

**Source data 2.** Original file for the Western blot analysis in *Figure 2G-L* (anti-pro-caspase3, anti-c-caspase3, anti-pro-caspase9, anti-c-caspase9, anti-Bax, anti-BCL-2, and anti-tubulin).

**Source data 3.** Original file for the Western blot analysis in *Figure 2M, N* (anti-PARP, anti-C-PARP, and anti-tubulin).

**Source data 4.** PDF containing *Figure 2A-N* and original scans of the relevant Western blot analysis (anti-p-Stat3, anti-Stat3, anti-MMP2, anti-PCNA, anti-Ki67, anti-Cyclin D1, anti-Cytochrome C, anti-pro-caspase3, anti-c-caspase3, anti-pro-caspase9, anti-c-caspase9, anti-Bax, anti-BCL-2, anti-PARP, anti-C-PARP, and anti-tubulin) with highlighted bands and sample labels.

Moreover, the combination treatment group demonstrated even greater inhibition of tumor growth than either mono-treatment group (*Figure 3B and C*). Notably, the combination treatment also significantly prolonged the survival of tumor-bearing mice, with three mice still surviving on the 60th day (*Figure 3D*). Histological analysis using HE staining showed that the tumor cells in the PBS group had an irregular shape, high nucleo-cytoplasmic ratio, a significant increase in heterotypic nuclei, and more pleomorphism of nuclei. However, the histomorphology significantly improved after the treatment

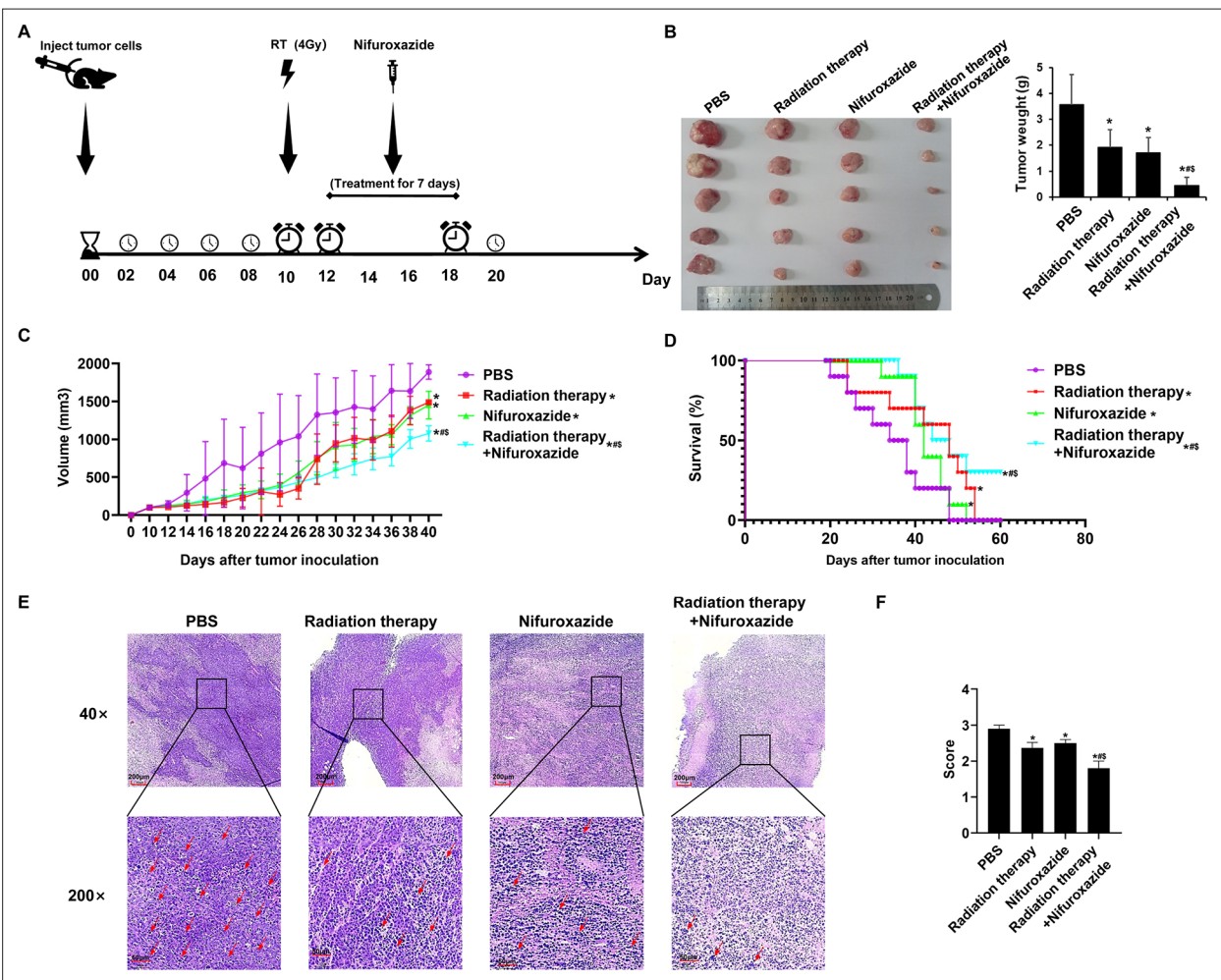

**Figure 3.** The effects of radiotherapy in combination with nifuroxazide on tumor growth and survival of tumor-bearing mice. At 7 days after establishing the tumor model, the mice are received distinct treatments. (**A**) Treatment scheme of different methods. (**B**) Tumor pictures and tumor weight of tumor-bearing mice under different treatments are measured for statistical analysis (n=5). (**C**) The tumor volume changes of tumor-bearing mice under different treatments (n=10). (**D**) The survival of tumor-bearing mice under different treatments (n=10). (**E**) Pathological pictures of tumor-bearing mice under different treatments. The ruler in the top row of the images represents 200 nanometers; the ruler in the bottom row of the images represents 50 nanometers. (**F**) Statistical analysis of pathological score (n=3). One-way analysis of variance (ANOVA) was carried out and the data are presented as mean ± SD except in Figure 3D, which was analyzed by the Kaplan-Meier method. Compared with the PBS group, *$p<0.05$; compared with the radiotherapy group, $$p<0.05$; compared with the nifuroxazide group, #$p<0.05$.

with nifuroxazide and radiation therapy (*Figure 3E*). Histological analysis using HE staining showed that, in the PBS group, tumor cells exhibited vigorous growth, tight arrangement, irregular shape, high nucleo-cytoplasmic ratio, significant increase in heterotypic nuclei, and more pleomorphism of nuclei. After treatment, the tumor cell arrangement became loose, and the cells displayed an irregular shape, with large vacuoles or vacancies, as well as necrosis or ablation phenomena. These changes were most significant in the combination treatment group, suggesting that the histomorphology significantly improved after the treatment with nifuroxazide and radiation therapy (*Figure 3E*). These findings indicated that nifuroxazide played a crucial role in enhancing the efficacy of radiotherapy for HCC.

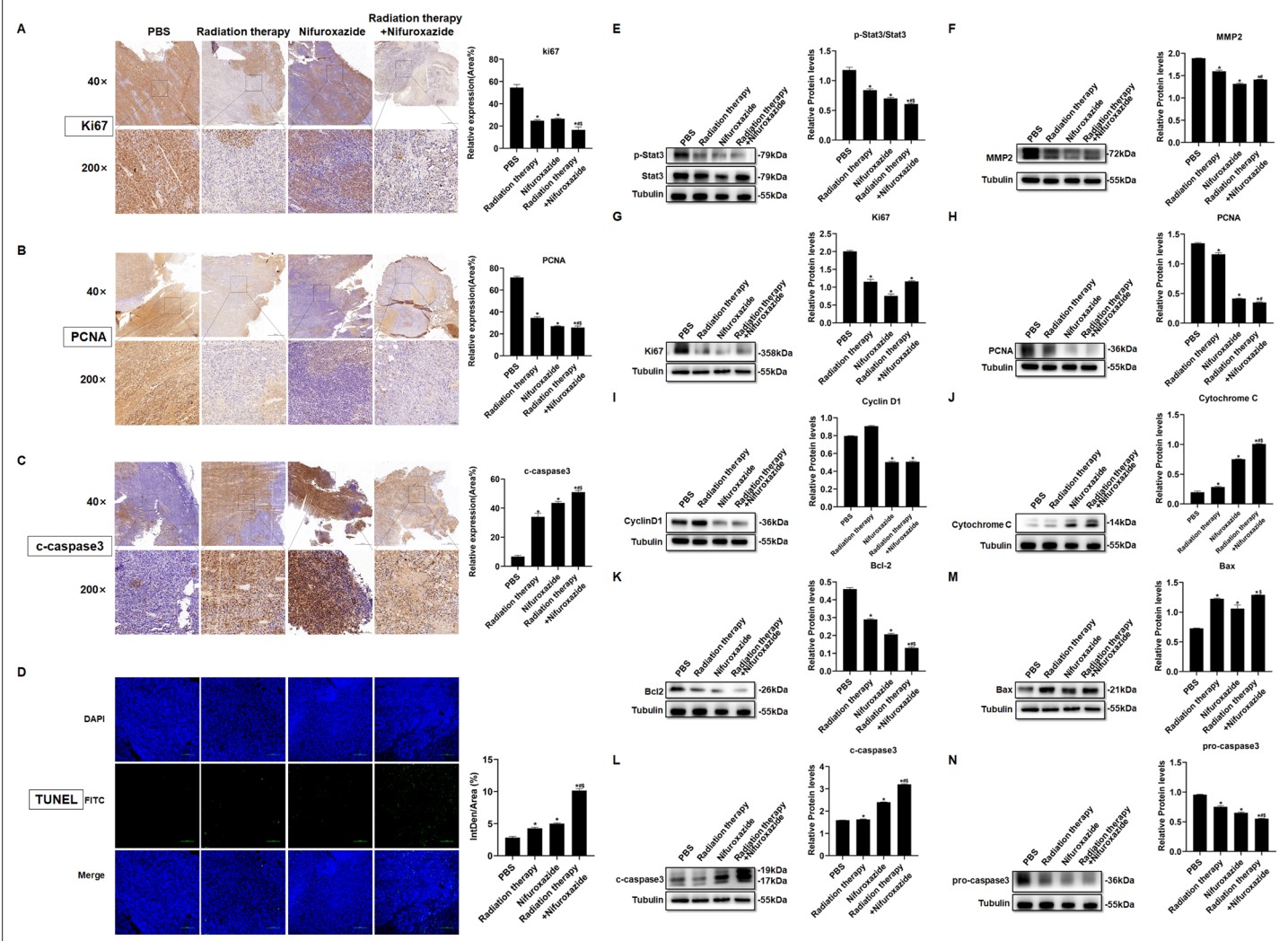

**Figure 4.** The effects of radiotherapy in combination with nifuroxazide on the cell proliferation or apoptosis in tumor tissues. (**A–C**) The expression of Ki6, PCNA, or c-caspase-3 in tumor tissues is detected by immunohistochemistry. The ruler in the top row of the images represents 200 nanometers; the ruler in the bottom row of the images represents 50 nanometers. (**D**) The cell apoptosis on tumor tissues is detected by TUNEL assay. (**E–N**) The protein expression of Stat3, p-Stat3, Ki67, PCNA, cyclin D1, cytochrome C, Bcl-2, Bax, pro-caspase 3, and c-caspase 3 in tumor tissues is detected by Western blot. One-way analysis of variance (ANOVA) was carried out and the data are presented as mean ± SD (n=3). Compared with the PBS group, *p<0.05; compared with the radiotherapy group, #p<0.05; compared with the nifuroxazide group, $p<0.05.

The online version of this article includes the following source data for figure 4:

**Source data 1.** Original file for the Western blot analysis in *Figure 4E–N* (anti-p-Stat3, anti-Stat3, anti-MMP2, anti-Ki67, anti-PCNA, anti-Cyclin D1, anti-Cytochrome C, anti-BCL-2, anti-Bax, anti-c-caspase3, anti-pro-caspase3, and anti-tubulin).

**Source data 2.** PDF containing *Figure 4E–N* and original scans of the relevant Western blot analysis (anti-p-Stat3, anti-Stat3, anti-MMP2, anti-Ki67, anti-PCNA, anti-Cyclin D1, anti-Cytochrome C, anti-BCL-2, anti-Bax, anti-c-caspase3, anti-pro-caspase3, and anti-tubulin) with highlighted bands and sample labels.

## The combination of radiotherapy and nifuroxazide showed a remarkable inhibition of cell proliferation and an increase in cell apoptosis in tumor tissues

To further elucidate the anti-tumor mechanism of combining radiotherapy with nifuroxazide, the expression of proteins related to proliferation and apoptosis was evaluated in tumor tissues of tumor-bearing mice. As shown in *Figure 4A–C*, the combination therapy significantly suppressed the expression of Ki67 and PCNA in tumor tissues while promoting the activation of caspase3, indicating its efficacy in hindering tumor cell proliferation and inducing apoptosis. Additionally, the TUNEL assay showed a marked increase in the apoptotic rate of tumor cells upon the combination therapy (*Figure 4D*). Moreover, the levels of proteins involved in cell proliferation (p-Stat3, PCNA, Ki67, and cyclin D1) and cell migration (MMP2) were analyzed in the tumor tissues. *Figure 4E–I* revealed that the combination therapy strikingly reduced the levels of associated proteins, indicating its potential to suppress tumor cell proliferation. Subsequently, the cell apoptosis in tumor tissues was evaluated, and the expression of pro-apoptotic proteins (c-caspase3, Bax, and Cytochrome C) was observed to be significantly upregulated in the tumor tissues of mice treated with radiotherapy plus nifuroxazide, whereas the expression of anti-apoptotic protein (BCL-2) was downregulated (*Figure 4J–N*). These results suggest that the combination therapy of radiotherapy and nifuroxazide effectively inhibits tumor cell proliferation and induces apoptosis by regulating the expression of proteins related to proliferation and apoptosis.

## The combination of radiotherapy and nifuroxazide significantly boosted the activation of tumor-infiltrating lymphocytes and increased the population of M1 macrophages in tumor-bearing mice

The impact of combining radiotherapy with nifuroxazide was comprehensively elucidated by assessing the infiltration of lymphocytes and M1 macrophages in tumor tissues. The combination therapy significantly augmented the infiltration of CD4$^+$ (*Figure 5A and E* and *Figure 5—figure supplement 1A*, confirmed by both immunofluorescence and flow cytometry) and CD8$^+$ (*Figure 5B and E* and *Figure 5—figure supplement 1B*, confirmed by both immunofluorescence and flow cytometry) lymphocytes, and stimulated the activation of lymphocytes (Granzyme B$^+$) (*Figure 5C and E* and *Figure 5—figure supplement 1C*, confirmed by both immunofluorescence and flow cytometry). Additionally, it increased the polarization of M1 macrophages (CD11$^+$ CD86$^+$) (*Figure 5D and E*). Moreover, the combination therapy increased the infiltration of NK cells (*Figure 5—figure supplement 1D*) and decreased the number of Treg cells (*Figure 5—figure supplement 1E*) in the cancer tissues of the HCC-bearing mice. Furthermore, the expression of associated proteins also demonstrated the same results (*Figure 5F–I*). These findings indicate that the combination of radiotherapy and nifuroxazide can effectively enhance the anti-tumor immune response in mice with tumors.

## Radiotherapy combined with nifuroxazide noticeably regulated the ratios of immune cells in the spleens

As the spleen is an essential immune organ, it plays a crucial role in the peripheral immune response. To quantify the ratios of immune cells in different groups of tumor-bearing mice, flow cytometry was employed. The results showed that treatment with radiotherapy or nifuroxazide led to a substantial increase in the number of CD8$^+$ lymphocytes, activated lymphocytes (Granzyme B$^+$), M1 macrophages, and NK cells in the spleens of mice compared to the PBS group. Moreover, the combined treatment resulted in the highest number of these cells among all the groups (*Figure 6B, D, E and F*). Additionally, the ratio of CD4$^+$ T lymphocytes in the spleens of mice treated with radiotherapy did not show any difference when compared to the PBS group. However, the ratio was significantly increased when radiotherapy was combined with nifuroxazide (*Figure 6A*). Furthermore, Treg cells are crucial immunosuppressive cells that facilitate the immune evasion of tumor cells. Our findings indicate that treatment with radiotherapy leads to a significant increase in the ratio of Treg cells (*Figure 6C*). However, the trend could be reversed after administering nifuroxazide treatment, suggesting that nifuroxazide may enhance the antitumor effects of radiotherapy by regulating the immune response.

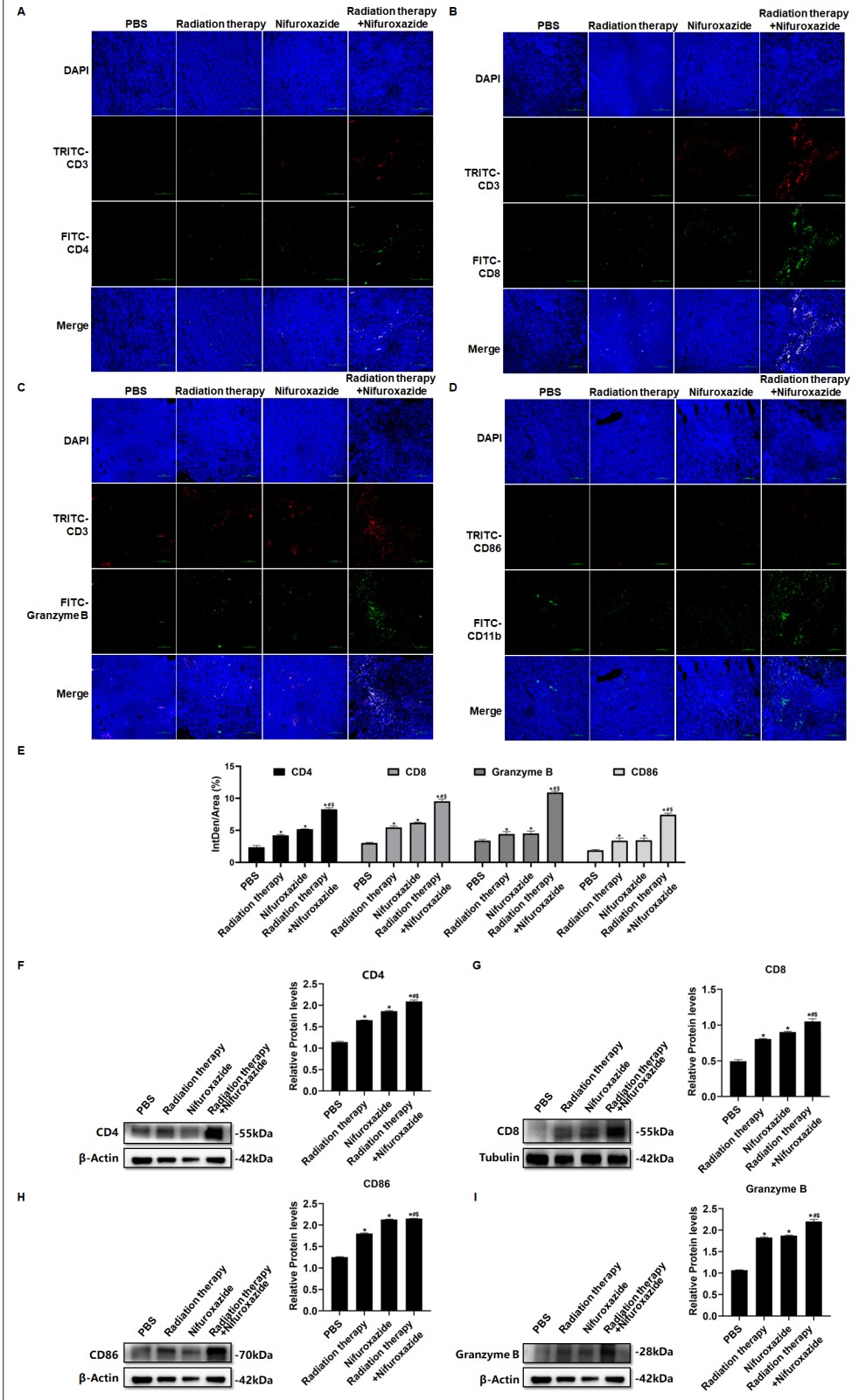

**Figure 5.** The effects of radiotherapy in combination with nifuroxazide on the infiltration of immune cells in tumor tissues. (**A–C**) Different isoforms of T lymphocytes infiltration in tumor tissues detected by immunofluorescence assay. (**D**) M1 macrophage infiltration in tumor tissues detected by immunofluorescence assay. (**E**) Statistical analysis about Semi-quantitative of Figure A-D. (**F–I**) The protein expression of CD4, CD8, CD86, and Granzyme in

*Figure 5 continued on next page*

*Figure 5 continued*

tumor tissues is detected by Western blot. One-way analysis of variance (ANOVA) was carried out and the data are presented as mean ± SD (n=3). Compared with the PBS group, *p<0.05; compared with the radiotherapy group, #p<0.05; compared with the nifuroxazide group, $p<0.05.

The online version of this article includes the following source data and figure supplement(s) for figure 5:

**Source data 1.** Original file for the Western blot analysis in *Figure 5F–I* (anti-p-CD4, anti-CD8, anti-CD86, anti-Granzyme B, and anti-tubulin).

**Source data 2.** PDF containing *Figure 5F–I* and original scans of the relevant Western blot analysis (anti-p-CD4, anti-CD8, anti-CD86, anti-Granzyme B, and anti-tubulin) with highlighted bands and sample labels.

**Figure supplement 1.** The effect of combined nifuroxazide with radiotherapy on the immunocyte in tumor tissue of tumor-bearing mice.

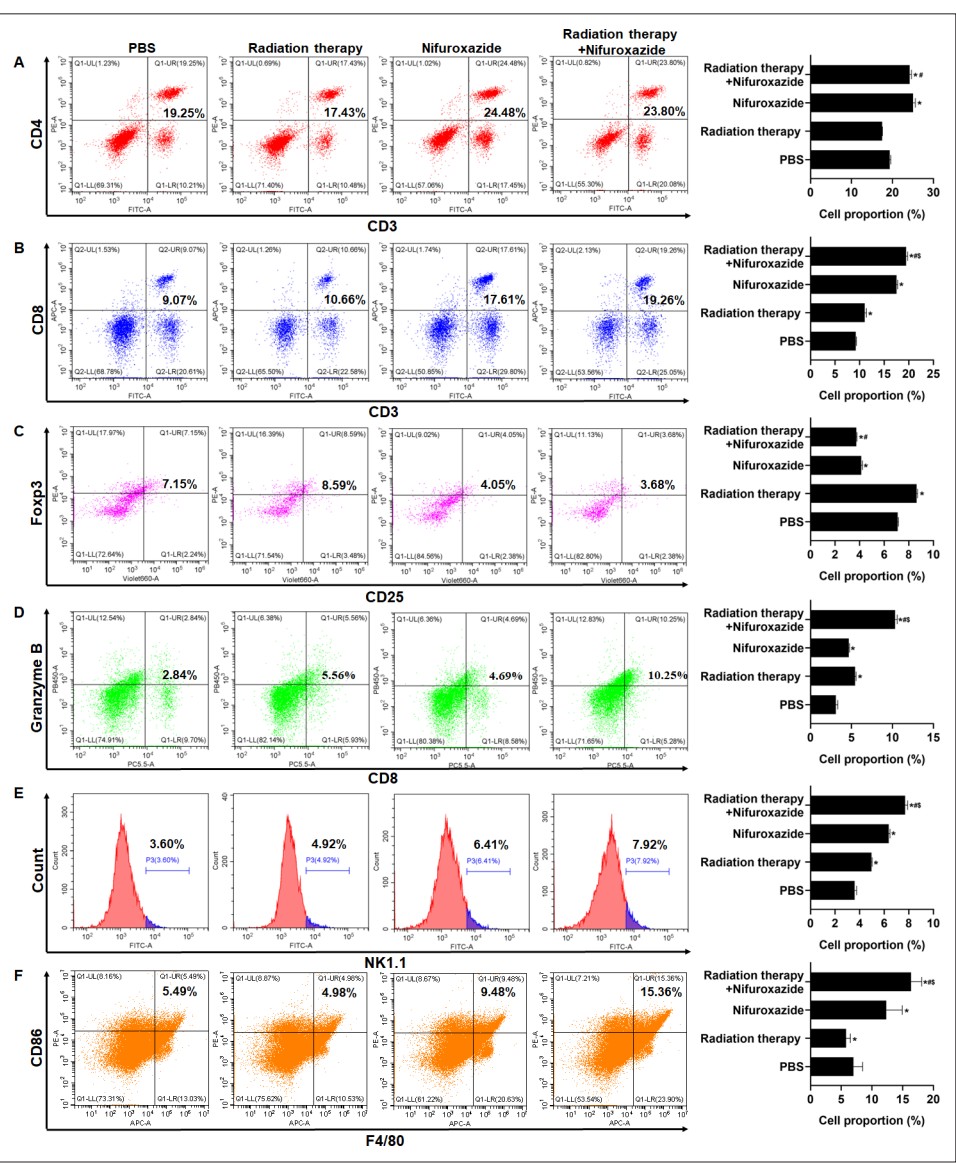

**Figure 6.** The effect of radiotherapy in combination with nifuroxazide on the ratios of immune cells in spleens. The ratios of immune cells in the spleens of tumor-bearing mice were detected by flow cytometry. One-way analysis of variance (ANOVA) was carried out and the data are presented as mean ± SD (n=3). Compared with the PBS group, *p<0.05; compared with the radiotherapy group, #p<0.05; compared with the nifuroxazide group, $p<0.05.

## Nifuroxazide in combination with the radiotherapy significantly increased the degradation of PD-L1 through the ubiquitin-proteasome pathway

PD-L1 is a co-inhibitory protein expressed by various cells, including tumor cells, and its pathway has been shown to inhibit the anti-tumor function of T lymphocytes. PD-L1 is considered a critical indicator of radiotherapy resistance. Therefore, we investigated whether nifuroxazide enhances radiosensitivity in HCC via the PD-1/PD-L1 pathway. Our results showed that radiotherapy significantly inhibited the expression of p-Stat3 (*Figure 2A and B*), but induced the upregulation of PD-L1 expression, which was effectively reversed by combined treatment with nifuroxazide (*Figure 7A* and *Figure 7—figure supplement 1A*, in both HepG2 and Huh7 cell lines). These results suggest that nifuroxazide may be a potential PD-L1 inhibitor, but its effect is independent of the Stat3 pathway. Surprisingly, we found that radiotherapy also upregulated PD-L1 mRNA expression, which was not affected by nifuroxazide treatment (*Figure 7B* and *Figure 7—figure supplement 1B*, in both HepG2 and Huh7 cell lines). These results suggest that nifuroxazide cannot regulate PD-L1 gene expression. The proteasome pathway is a key pathway involved in protein degradation. Our findings demonstrate that the inhibitory effect of nifuroxazide on PD-L1 expression was counteracted by the proteasome inhibitor (*Figure 7C* and *Figure 7—figure supplement 1C*, in both HepG2 and Huh7 cell lines). Moreover, nifuroxazide treatment clearly increased the expression of GSK3β (*Figure 7D*). Additionally, the inhibitor of GSK3β reversed the downregulation of *CD274* expression in cells treated with nifuroxazide (*Figure 7E*). These findings confirmed that nifuroxazide could increase the degradation of PD-L1 via the ubiquitin-proteasome pathway and ultimately restore the immunity of T lymphocytes (*Figure 7F*).

## The combination of nifuroxazide and radiotherapy notably inhibited the expression of PD-L1 via the up-regulation of GSK3β expression

We next detected the expression level of PD-L1 in tumor tissues of HCC patients pre- and post-radiotherapy. The results showed a significant increase in PD-L1 expression in tumor tissues of HCC patients after radiotherapy, compared to pre-radiotherapy (*Figure 8A*), which may account for the poor clinical response to radiotherapy. Similarly, we observed an elevation in PD-L1 expression in tumor tissues of mice receiving radiotherapy. However, treatment with nifuroxazide effectively blocked the up-regulation of PD-L1 induced by radiotherapy (*Figure 8B and C*). Notably, the study demonstrated that treatment with nifuroxazide significantly increased the expression of GSK3β in mice, regardless of radiotherapy treatment (*Figure 8D*). These findings suggest that nifuroxazide may enhance the degradation of PD-L1 and improve the therapeutic response to radiotherapy.

## Discussion

Radiation therapy is a crucial approach in the management of HCC, but its effectiveness is limited by radioresistance. Our investigation has confirmed that nifuroxazide can enhance the immune response against tumors induced by radiation therapy by promoting the degradation of PD-L1 in HCC. PD-L1 is predominantly expressed on the surface of tumor cells and severely obstructs the immune response against tumors by binding to PD-1. Several studies have indicated that overexpression of PD-L1 is a major cause of radioresistance (*Boustani et al., 2021*; *Yi et al., 2022*; *Gong et al., 2018*). Therefore, inhibition of PD-L1 may prove to be an effective strategy to strengthen the antitumor effect of radiation therapy. D-mannose has been reported to significantly enhance the effects of radiotherapy by promoting PD-L1 degradation in triple-negative breast tumors (*Zhang et al., 2022*). Additionally, Stat3 plays a key role in tumor progression and also regulates PD-L1 expression (*Tong et al., 2020*). Therefore, investigating Stat3 inhibitors could offer a fresh perspective on suppressing PD-L1 expression. A new Stat3 inhibitor was discovered, which is nifuroxazide, previously used to treat diarrhea (*Nelson et al., 2008*). Our study showed that nifuroxazide significantly inhibited the protein expression of PD-L1. Further investigations demonstrated that radiation therapy upregulated the gene level of PD-L1, while nifuroxazide did not change the gene level in vitro. Intriguingly, nifuroxazide covertly promoted the expression of GSK3β in cells that have undergone radiography. Simultaneously, the protease inhibitor blocked the inhibitory effect on PD-L1 expression. This finding suggests that nifuroxazide modulates the PD-L1 pathway through the ubiquitin-proteasome pathway rather than the mRNA pathway.

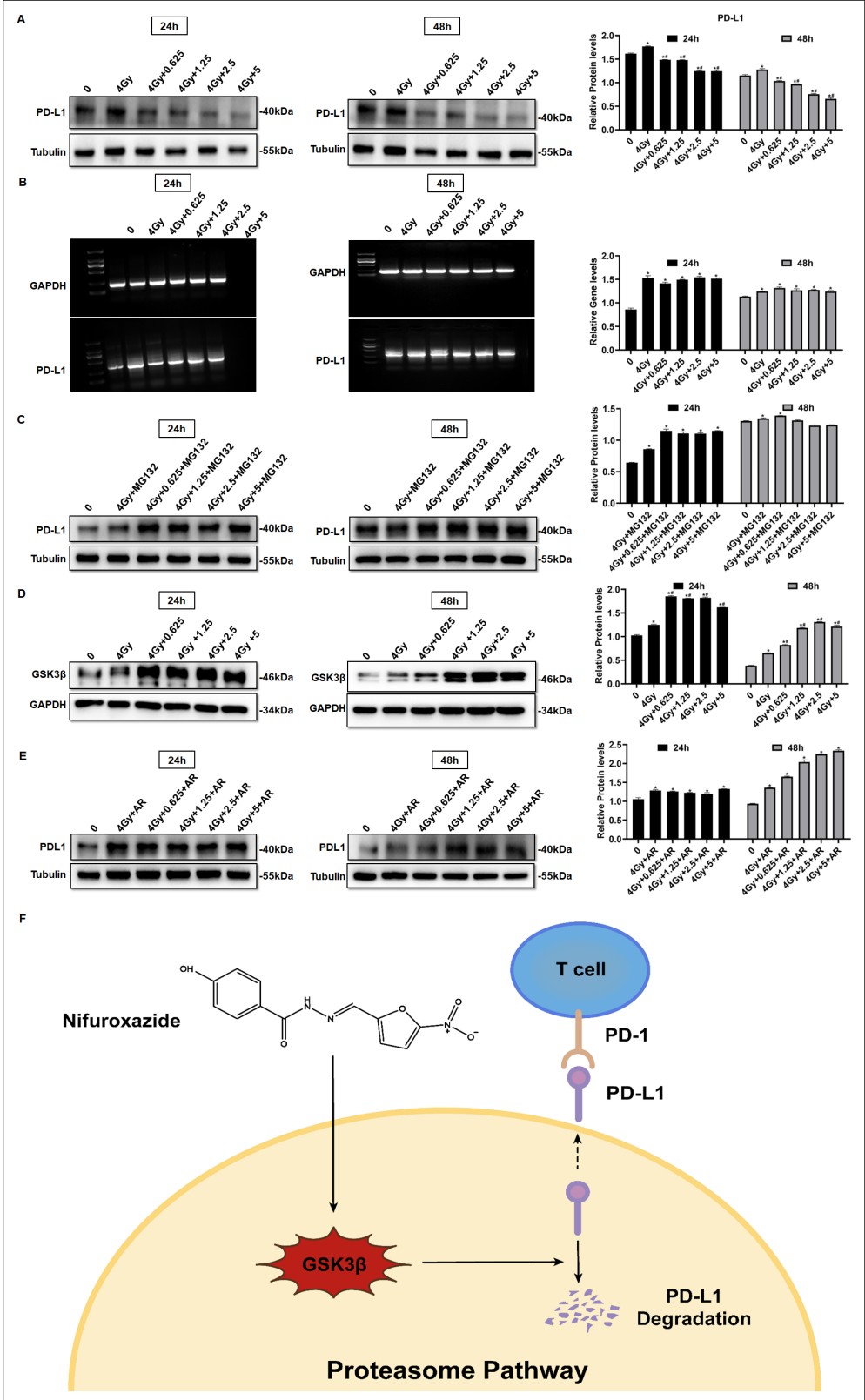

**Figure 7.** Nifuroxazide in combination with the radiotherapy significantly increases the degradation of pd-l1 through the ubiquitination proteasome pathway. (**A**) The expression of PD-L1 in cells combined treatment with radiotherapy and nifuroxazide is detected by Western blot. (**B**) The mRNA level of PD-L1 in cells is detected by PCR. (**C–E**) The effect of the radiotherapy combined with nifuroxazide on PD-L1 degradation through the

*Figure 7 continued on next page*

*Figure 7 continued*

ubiquitination-proteasome pathway. (**F**) The schematic diagram of the mechanism that nifuroxazide degraded PD-L1 through the ubiquitination-proteasome pathway. One-way analysis of variance (ANOVA) was carried out and the data are presented as mean ± SD (n=3). Compared with '0' group, $*p<0.05$; compared with '4 Gy' group, $\#p<0.05$.

The online version of this article includes the following source data and figure supplement(s) for figure 7:

**Source data 1.** Original file for the Western blot analysis in *Figure 7A and C–E* (anti-PD-L1, anti-GSK3β, and anti-tubulin).

**Source data 2.** PDF containing *Figure 7A and C–E* and original scans of the relevant Western blot analysis (anti-PD-L1, anti-GSK3β, and anti-tubulin) with highlighted bands and sample labels.

**Figure supplement 1.** The effect of nifuroxazide on PD-L1 expression in Huh7 cells.

**Figure supplement 1—source data 1.** Original file for the Western blot analysis in *Figure 7—figure supplement 1A and C* (anti-PD-L1, and anti-tubulin).

**Figure supplement 1—source data 2.** PDF containing *Figure 7—figure supplement 1A and C* and original scans of the relevant Western blot analysis (anti-PD-L1, and anti-tubulin) with highlighted bands and sample labels.

The anti-tumor role of nifuroxazide has been reported. However, the innovation of our study does not lie in confirming its antitumor effects but rather in demonstrating how nifuroxazide can enhance radiotherapy's efficacy in treating hepatocellular carcinoma by inhibiting PD-L1 levels. We compared the efficacy of the combination therapy versus radiotherapy and found that compared to radiotherapy alone, the combination therapy more significantly inhibited hepatocellular carcinoma cell proliferation and migration. In our animal model, we compared the therapeutic effects of the combination therapy, nifuroxazide alone, and radiotherapy alone on hepatocellular carcinoma-bearing mice. We observed that compared to individual treatment groups, the combination therapy more profoundly suppressed tumor growth and enhanced the antitumor effects in the mice.

The primary objective of this study is to explore whether nifuroxazide can effectively enhance the degradation of PD-L1, thereby increasing the radiosensitivity of HCC. Our research reveals that compared to radiation therapy alone, combination therapy involving nifuroxazide and radiation significantly inhibits tumor growth in mice and boosts the anti-tumor immune response. This finding could potentially provide a valuable strategy for patients who exhibit resistance to radiation therapy in clinical practice. Moreover, clinical trial investigations have demonstrated that nivolumab, a PD-1 monoclonal antibody, when combined with radiotherapy for HCC, exhibits promising safety and efficacy (*de la Torre-Aláez et al., 2022*). This evidence supports the future application of nifuroxazide in the treatment of HCC. However, to reach this objective, we must continue to conduct extensive research, including comparing nifuroxazide with existing therapies in clinical practice. In this study, nifuroxazide not only significantly inhibits the expression of PD-L1 protein in HCC cells but also functions as a PD-L1 inhibitor. Furthermore, it effectively curbs the proliferation and migration of HCC cells, induces tumor cell apoptosis, and may exhibit enhanced anti-tumor effects, making it a promising candidate for clinical use. We believe that the advantage of nifuroxazide over PD-1 or PD-L1 antibodies lies in its ability not only to effectively inhibit PD-L1 expression but also to suppress tumor cell proliferation, migration, and promote tumor cell apoptosis.

We analyzed the underlying anti-tumor mechanism of nifuroxazide in terms of cell proliferation, migration, and drug action target. Research has shown that regulating the STAT3/PD-L1 pathway can effectively increase apoptosis in lung cancer cells (*Xie et al., 2021*). Our study confirmed that nifuroxazide can effectively inhibit the expression of p-STAT3 and PD-L1 in liver cancer cells, which may be the reason for the increased apoptosis of these cells. Nifuroxazide has been demonstrated to inhibit the expression of p-STAT3, thereby suppressing tumor cell proliferation and migration (*Nelson et al., 2008*; *Yang et al., 2015*). In our study, we observed that after 48 hr of treatment with nifuroxazide, the expression of p-STAT3 in irradiated cells was significantly inhibited. Furthermore, compared to radiotherapy alone, combining nifuroxazide and radiotherapy resulted in a more pronounced decrease in PCNA expression. Simultaneously, we performed additional detection of migration-related protein MMP2, confirming that combining nifuroxazide and radiotherapy led to a more significant inhibition of MMP2 expression. These findings suggest that the combination treatment may be responsible for the synergistic suppression of HCC cell proliferation and migration. Our initial results indicate that nifuroxazide inhibits the expression of PD-L1 at the protein level, but does not affect its mRNA level.

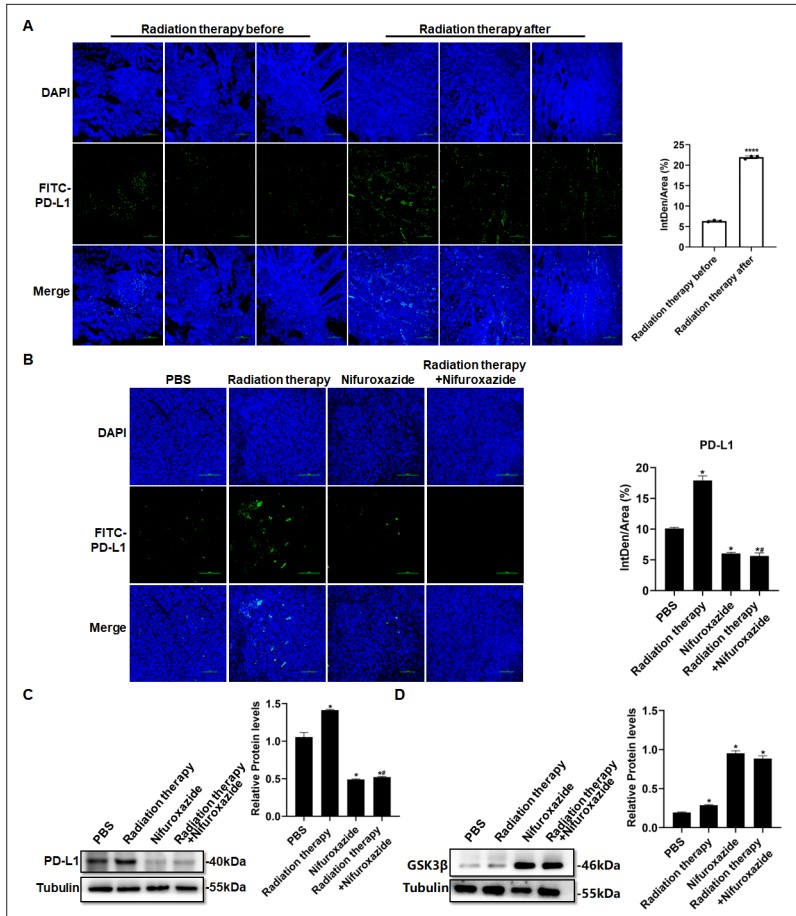

**Figure 8.** The effect of radiotherapy in combination with nifuroxazide on the expression of PD-L1 in tumor tissues. (**A**) PD-L1 expression in tumor tissues of hepatocellular carcinoma (HCC) patients treated with radiotherapy are detected by immunofluorescence. (**B**) PD-L1 expression in tumor tissues of mice combined treatment with radiotherapy and nifuroxazide is detected by immunofluorescence. (**C–D**) The expression of PD-L1 or GSK3β in tumor tissues of mice combined treatment with radiotherapy and nifuroxazide is detected by Western blot. One-way analysis of variance (ANOVA) was carried out and the data were expressed as mean ± SD (n=3). Compared with the before-radiotherapy samples, ****$p<0.0001$; compared with the PBS group, *$p<0.05$; compared with the radiotherapy group, #$p<0.05$; compared with the nifuroxazide group, $$p<0.05$.

The online version of this article includes the following source data for figure 8:

**Source data 1.** Original file for the Western blot analysis in *Figure 8C–D* (anti-PD-L1, anti-GSK3β, and anti-tubulin).

**Source data 2.** PDF containing *Figure 8C–D* and original scans of the relevant Western blot analysis (anti-PD-L1, anti-GSK3β, and anti-tubulin) with highlighted bands and sample labels.

Interestingly, upon treatment with a proteasome inhibitor MG132, the inhibitory effect of nifuroxazide on PD-L1 was eliminated, suggesting that nifuroxazide may enhance the degradation of PD-L1 protein. Our experiments have demonstrated the inhibitory effect of Nifuroxazide on PD-L1 in both human and mouse cell lines.

Furthermore, the overexpression expression of PD-L1 hinders the antitumor immunity response by inhibiting T lymphocyte function and macrophage polarization (*Cousin et al., 2021*; *Xiong et al., 2019*). The activity of T lymphocytes is closely associated with tumor progression, but it can become dysregulated due to the interaction between PD-1 and PD-L1 (*Sun et al., 2018*; *Dammeijer et al., 2020*). Blocking the PD-1 and PD-L1 pathways can reverse T-cell exhaustion and enhance their ability to eliminate tumor cells (*Budimir et al., 2022*). In our study, treatment with nifuroxazide effectively boosted T-cell activation, with a more pronounced effect observed in mice receiving radiation therapy. Radiation therapy has been shown to induce an immunogenic effect that promotes T lymphocyte

infiltration into tumors (*Lhuillier et al., 2019*), but it also induces PD-L1 expression, which impairs T lymphocyte function (*Lan et al., 2021*). Under these circumstances, nifuroxazide significantly enhanced T lymphocyte activation in mice receiving radiation therapy by reducing PD-L1 expression.

Moreover, the upregulation of PD-L1 on the surface of macrophages has been found to promote the reduction of M1 macrophage polarization, which is recruited by tumor cells, leading to the inhibition of antitumor immunity (*Hartley et al., 2018*; *Ubil et al., 2018*). Studies have indicated a distinct decline in M1 macrophages in tumor tissues of patients undergoing radiation therapy (*Brown et al., 2020*; *Yang et al., 2020*). Therefore, a crucial strategy to enhance the antitumor effect of radiation therapy is to reverse macrophage polarization. Our research confirmed that nifuroxazide had a positive effect on promoting M1 macrophage polarization. Notably, while radiation therapy suppressed the infiltration of M1 macrophages, the number of M1 macrophages significantly increased in tumor tissues of mice treated with nifuroxazide and radiation therapy. These findings demonstrate that nifuroxazide plays a vital role in strengthening the antitumor immunity of radiation therapy for HCC by reversing macrophage polarization.

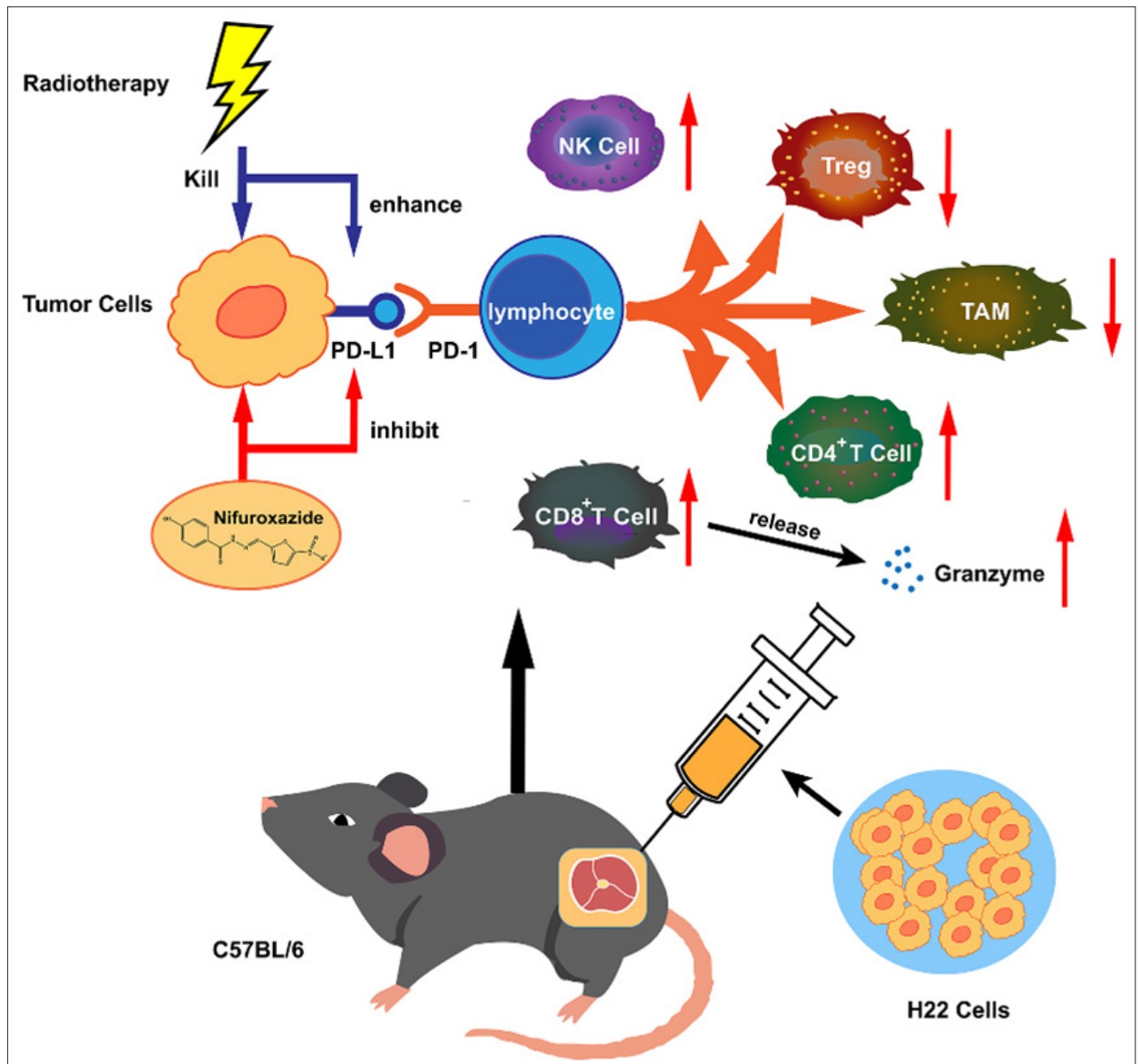

**Figure 9.** The synergistic antitumor mechanism of radiotherapy in combination with nifuroxazide in tumor-bearing mice. NK cell: natural killer cell. Treg: regulatory T cells. TAM: tumor-associated macrophage. CD8[+] CTL: CD8[+] cytotoxic lymphocyte.

To summarize, the antitumor effect of radiation therapy is impaired due to the induction of PD-L1 expression. High expression of PD-L1 hinders the antitumor function of T lymphocytes and macrophages. However, nifuroxazide significantly inhibits PD-L1 up-regulation induced by radiation therapy, enhances the activation of T lymphocytes and the ratio of M1 macrophages, and decreases the number of Treg cells (*Figure 9*). We confirm that nifuroxazide improves the antitumor effect of radiotherapy and provides a synergistic therapeutic strategy for HCC patients, especially those who are radioresistant. However, to achieve the goal of applying nifuroxazide combined with radiation therapy for the treatment of clinical hepatocellular carcinoma, we still need to undertake extensive research, including validation on genetically identical mouse HCC tumor models.

## Materials and methods

### Regents and cell lines

The Nifuroxazide used in this study was obtained from Sigma-Aldrich (USA). The HepG2, Huh7, and H22 cell lines were maintained at Xinxiang Key Laboratory of Tumor Vaccine and Immunotherapy, located at Xinxiang Medical University in Xinxiang, Henan (P.R. China).

### Cell viability assay

HepG2 and Huh7 cells were cultured and seeded in 96-well plates (Corning, New York, NY) at a density of $1 \times 10^4$/well. After 15 hr of incubation, nifuroxazide with varying concentrations (0, 0.3125, 0.625, 1.25, 2.5, 5, 10, 20, 40 µg/mL) were added and co-incubated for 24 or 48 hr with or without radiotherapy treatment. After the incubation period, Cell Counting Kit-8 (CCK-8, Beyotime Institute of Biotechnology, Shanghai, China) reagents were added into each well and incubated for an additional 2 hr. The optical density (OD) was measured at 450 nm using a microplate reader (SpectraMax iD3, Molecular Devices).

### Wound-healing assay

HepG2 and Huh7 cells were cultured and subjected to radiotherapy, then seeded in six-well plates (Corning, New York, NY) at a density of $3.5 \times 10^5$ cells per well, followed by an incubation period of 15 hr. A scratch was created in the middle of each well using a pipette tip. Afterwards, the cells were co-cultured with varying concentrations of nifuroxazide (0, 0.625, 1.25, 2.5, 5 µg/mL) for a period of 24 or 48 hr. Subsequently, the width of the scratch was recorded and measured.

### Colony formation

HepG2 cells were treated with or without radiotherapy and then plated at a density of $2 \times 10^3$ cells per well in six-well plates from Corning, New York, NY. After an incubation period of 15 hr, nifuroxazide was added at various concentrations (0, 0.625, 1.25, 2.5, 5 µg/mL). The culture medium was periodically replaced and cell masses became visible by the 10th day. Subsequently, the cell colonies were detected and recorded.

### Transwell

HepG2 cells were treated with or without radiotherapy and then plated at a density of $1 \times 10^5$ cells per well in the upper chamber, while different concentrations of nifuroxazide (0, 0.625, 1.25, 2.5, 5 µg/mL) were added. Meanwhile, 500 µL of culture medium was added to the lower chamber. After a 24 hr incubation period, the lower chambers were drained and stained with crystal violet for 40 min. The results were subsequently observed under a microscope (Nikon, Japan) after sealing with neutral resin.

### RT-PCR

HepG2 cells, treated with or without radiotherapy, were seeded at a density of $3.5 \times 10^5$ cells per well in six-well plates from Corning, New York, NY, and incubated for 15 hr. Nifuroxazide was then added at varying concentrations (0, 0.625, 1.25, 2.5, 5 µg/mL) into each well. After a period of 24 or 48 hr, total RNA was extracted from the cells using the Trizol reagent, following the instructions. The gene expression of *CD274* was subsequently detected using PCR. The PCR primers used were as follows: For GAPDH, Forward-sequence: AGAAGGCTGGGGCTCATTTG, and Reverse-sequence: AGGG

GCCATCCACAGTCTTC. For PD-L1, Forward-sequence: GAGCGTGACAAGAGGAAGGAATGG, and Reverse-sequence: TTGAGGCATTGAGTGGAGGCAAAG.

### Detection of mRNA levels

Calculate and perform statistical analysis on the gray values of the agarose gel electrophoresis bands. Divide the gray value of the target gene by the gray value of the internal control to correct for errors. The resulting value represents the relative content of the target gene in a specific sample.

### Establishment of tumor model and therapeutic options

The female C57BL/6 mice with 6–8 weeks were procured from Skbex Biotechnology in Henan, China. The animal studies were approved by the Ethics Committee of Xinxiang Medical University located in Xinxiang, China. H22 cells were subcutaneously injected into the right dorsal of the mice at a concentration of $2×10^6/100$ μl. After 7 days, the mice were randomly assigned to 1 of 4 groups: the PBS group, Radiation therapy group, nifuroxazide group, and Radiation therapy plus Nifuroxazide group. Seven days after the inoculation, the mice were anesthetized using 1% sodium pentobarbital and underwent radiation therapy at a dose of 4 Gy. The following day, nifuroxazide was intratumorally injected at a dose of 200 μg per mouse once a day. On the eighth day after the last treatment, relevant indicators were measured. The number of mice for the survival rate is 10 per group, while the number of mice for the other experiment is six per group.

### Western blot

Total proteins were extracted using RIPA lysate buffer (Beyotime Institute of Biotechnology, Shanghai, China) for western blot detection. Briefly, the proteins were separated via 10% sodium dodecyl sulfate–polyacrylamide gel electrophoresis (SDS–PAGE). Following electrophoresis, the proteins were transferred to a polyvinylidene fluoride membrane (EMD Millipore, Billerica, MA, USA). The PVDF membranes were then incubated with a buffer of 5% nonfat dry milk for 2 hr at room temperature. After being washed, the membranes were incubated with the following primary antibodies overnight at 4°C: PD-L1 (1:1000, Bioworld), p-Stat3 (1:1000, CST), Stat3 (1:1000, Bioworld), cleaved-caspase 3 (1:1000, Bioworld), Pro-Caspase 3 (1:2000, Bioworld), MMP2 (1:1000, CST), Ki67 (1:1000, Bioworld), PCNA (1:1000, SANTA), GSK3β (1:2000, Abways), cyclin D1 (1:2000, SANTA), Bcl-2 (1:2000, Abways), Bax (1:2000, Abways), Cytochrome C (1:1000, Biopple), cleaved-caspase 9 (1:2000, Abways), Pro-Caspase 9 (1:2000, Abways), PARP (1:1000, CST), cleaved-PARP (1:1000, CST), and Tubulin (1:1000, Sigma). The membranes were then washed and incubated with secondary antibodies (ZSGB-BIO, 1:5000). Finally, specific protein bands were visible using enhanced chemiluminescence (Beyotime Institute of Biotechnology) and detected using the chemiluminescence imaging instrument (Fusion FX spectra, Vilber). The images were semi-quantified using Quantity One software (Version 4.62; Bio-Rad Laboratories, Inc, Hercules, CA, USA).

### Hematoxylin and eosin (HE) staining

Tumor tissue waxes were utilized to prepare tissue sections with a thickness of 4 μm. Following dewaxing and dehydration, the sections were stained using hematoxylin-eosin (Beyotime Biotechnology, Shanghai, China). Subsequently, the sections were rinsed and dried, and the resulting images were observed utilizing a microscope (Nikon, Japan). The tumor tissue is graded according to histological standards, including assessment of irregular tumor cell nuclei, nuclear abnormalities, and nuclear-to-cytoplasmic ratio. The scoring system for abnormal nuclei and nuclear-to-cytoplasmic ratio is as follows: 10 to 20%%=1 point, 20 to 30%%=2 points, and greater than 30%=3 points.

### Immunohistochemistry (IHC)

Following preparation as previously described (*Zhao et al., 2019*), the tissue sections were incubated with primary antibodies against Ki67 and cleaved-caspase3, both purchased from Cell Signaling Technology (USA), overnight at 4°C. Subsequently, the sections were washed and incubated with biotin-labeled secondary antibodies for 30 min. After the completion of the reaction, streptavidin labeled with horseradish peroxidase was added. After a 30 min incubation, the sections were chromogenically stained using the DAB reagent. Finally, the results were recorded utilizing a digital scanner (3DHIS-TECH, Pannoramic MIDI, China).

## Immunofluorescence (IF)

The tissue samples were incubated with primary antibodies (CD3, 1:100, OmnimAbs; CD4, 1:200, CST; CD8, 1:800, CST; Granzyme B, 1:200, Bioworld; CD86, 1:400, Novus; CD11b, 1:200, Abways; PD-L1, 1:200, Bioworld) overnight at 4°C. Subsequently, the samples were rinsed and exposed to secondary antibodies labeled with corresponding fluorescent markers (Abways). After incubating at room temperature for 30 min, the samples were stained with DAPI solution (Beyotime) and incubated for 5 min at room temperature. Lastly, the samples were washed and sealed with an anti-fluorescence quenching agent, and the images were captured using a confocal microscope (AR1+, Nikon).

## TUNEL

The detection of cell apoptosis in tumor tissues was conducted via TUNEL assay in accordance with the instructions provided in the manual from Beyotime Biotechnology. In brief, tumor sections were exposed to TUNEL detection solution and incubated at 37°C for 60 min. Following a wash step, the sections were incubated with an anti-fluorescence quenching agent. The resulting images were captured using a confocal microscope (AR1+, Nikon).

## Flow cytometry

Spleen cell suspensions were prepared and the removal of red blood cells was accomplished with the use of an erythrocyte lysate solution (Beyotime Biotechnology). The concentration of cells was then adjusted to $1 \times 10^7$ cells/ml. Subsequently, 100 µL of cell suspension was separately incubated with CD3, CD4, CD8, CD25, Foxp3, NK1.1, CD86, F4/80, and Granzyme B antibodies (Biolegend) for 30 min. The cells were then washed using PBS buffer, and the ratios of immune cells were assessed utilizing flow cytometry (Cyto FLEX, Beckman).

## Statistical analysis

Measurement data are expressed as the mean ± SD of three independent experiments, obtained from three separate and independent experimental trials. Statistical analysis was conducted using SPSS version 19.0, developed by IBM Corporation. Multigroup comparisons of the means were carried out by one-way analysis of variance (ANOVA) test with post hoc contrasts by the Student–Newman–Keuls test, and the survival rate was analyzed using the Kaplan-Meier method with a log-rank test. Results with p-values below 0.05 were deemed to be statistically significant.

## Acknowledgements

This study was financially supported by the Doctor Launch Fund of Xinxiang Medical University (grant nos. 505017, 502006 and 505016), and the Key Projects of Scientific Research for Higher Education of Henan Province (grant no. 21A310012), the Young Backbone Teacher Training Projects of Universities in Henan province (grant no. 2020GGJS149), the Science and Technology Research Project of Henan Province (grant no. 222102310016), the Major Science and Technology Project of Xinxiang City (ZD2020005), the Science and Technology Project of Xinxiang (GG2019043), the Medical Education Research Project of Henan Province (grant no. Wjlx2020305), the Teaching and Scientific Research Program of Basic Medical College, Xinxiang Medical University (grant no. JCYXYJX202006), the Graduate Student Innovation Support Plan (grant no. YJSCX202149Y), the Innovation and Entrepreneurship Training Programmed for college student (grant no. 202210472005), Key Research and Development Projects in Henan Province (grant no. 231111311300).

## Additional information

### Funding

| Funder | Grant reference number | Author |
|---|---|---|
| Doctor Launch Fund of Xinxiang Medical University | 505017 | Huijie Jia |

| Funder | Grant reference number | Author |
|---|---|---|
| Doctor Launch Fund of Xinxiang Medical University | 505016 | Tiesuo Zhao |
| Doctor Launch Found of Xinxiang Medical University | 502006 | Zhiwei Feng |
| Key Projects of Scientific Research for Higher Education of Henan Province | 21A310012 | Tiesuo Zhao |
| Young Backbone Teacher Training Projects of Universities in Henan Province | 2020GGJS149 | Tiesuo Zhao |
| Science and Technology Research Project of Henan Province | 222102310016 | Huijie Jia |
| Major Science and Technology Project of Xinxiang City | ZD2020005 | Feng Ren |
| Science and Technology Project of Xinxiang | GG2019043 | Yongxi Zhang |
| Medical Education Research Project of Henan Province | Wjlx2020305 | Tiesuo Zhao |
| Teaching and Scientific Research Program of Basic Medical College, Xinxiang Medical University | JCYXYJX202006 | Tiesuo Zhao |
| Graduate Student Innovation Support Plan | YJSCX202149Y | Pengkun Wei |
| Innovation and Entrepreneurship Training Program for College Student | 202210472005 | Shijie Zhou |
| Key Research and Development Projects in Henan Province | 231111311300 | Feng Ren |

The funders had no role in study design, data collection and interpretation, or the decision to submit the work for publication.

#### Author contributions

Tiesuo Zhao, Conceptualization, Resources, Supervision, Funding acquisition, Validation, Writing - original draft, Project administration; Pengkun Wei, Data curation, Software, Formal analysis, Investigation, Visualization, Methodology, Project administration; Congli Zhang, Investigation, Methodology; Shijie Zhou, Yongxi Zhang, Investigation, Project administration; Lirui Liang, Shuoshuo Guo, Sichang Cheng, Zerui Gan, Yuanling Xia, Sheng Guo, Jiateng Zhong, Zishan Yang, Fei Tu, Investigation; Zhinan Yin, Qianqing Wang, Jin Bai, Supervision; Feng Ren, Zhiwei Feng, Supervision, Funding acquisition, Project administration; Huijie Jia, Supervision, Funding acquisition, Validation, Project administration, Writing – review and editing

#### Author ORCIDs

Pengkun Wei http://orcid.org/0000-0002-7818-3068
Feng Ren https://orcid.org/0000-0002-9457-5168
Huijie Jia https://orcid.org/0000-0002-4267-0198

#### Ethics

This study was performed in accordance with the national guidelines, and with the approval of the Institution for the Care and Use of Animals (IACUC) and the Ethics Committee of Xinxiang Medical

University (Permit Number: XYLL-20220117). All surgery was performed under sodium pentobarbital anesthesia, and every effort was made to minimize suffering.

Reviewer #1 (Public Review): https://doi.org/10.7554/eLife.90911.3.sa1
Reviewer #2 (Public Review): https://doi.org/10.7554/eLife.90911.3.sa2
Reviewer #3 (Public Review): https://doi.org/10.7554/eLife.90911.3.sa3
Author Response https://doi.org/10.7554/eLife.90911.3.sa4

---

# Additional files

## Supplementary files
- Supplementary file 1. The list of antibodies used for cytometry in this study.
- Supplementary file 2. The list of antibodies used for western blotting in this study.
- MDAR checklist

## Data availability
All data needed to evaluate the conclusions in the paper are present in the article or the supporting files.

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
