## [Editor Report · eLife assessment]

This **valuable** study evaluates the effects of nifuroxazide on radiotherapy for the treatment of hepatocellular carcinoma. **Solid** evidence is provided to support the conclusion that nifuroxazide facilitates the downregulation of PD-L1 and may improve therapy outcomes when combined with radiotherapy, though the inclusion of additional cell lines and animal models would have strengthened the study. This work will be of interest to cancer biologists and those working in immuno-oncology.

---

## [Referee Report · Reviewer #1 (Public Review)]

The author found the nifuroxazide has the potential to augment the efficacy of radiotherapy in HCC by reducing PD-L1 expression. This effect may be attributed to increased degradation of PD-L1 through the ubiquitination-proteasome pathway. These evidences support the future application of nifuroxazide in the treatment of HCC.

---

## [Referee Report · Reviewer #2 (Public Review)]

Summary:

Zhao et al. aimed to explore an important question-how to overcome resistance of hepatocellular carcinoma cells to radiotherapy. Given that immune-suppressive microenvironment is a major mechanism underlying resistance to radiotherapy, they reasoned that a drug that blocks PD-1/PD-L1 pathway could improve efficacy of radiation therapy and chose to investigate the effect of Nifuroxazide, an inhibitor of stat3 activation, on radiotherapy efficacy in treating hepatocellular carcinoma cells. From in vitro experiments, they find combination treatment (Nifuroxazide+ radiotherapy) increases apoptosis and reduces proliferation and migration, in comparison to radiotherapy alone. From in vivo experiments, they demonstrate that combined treatment reduces size and weight of tumors in vivo and enhances mice survival. These data indicate a better efficacy of combination therapy compared to radiotherapy alone. Moreover, they also determined the effect of combination therapy on tumor microenvironment as well as peripheral immune response. Specifically, they find that combination therapy increases infiltration of CD4+, CD8+ t cells and NK cells, activates CD8+ t cells, enhances polarization of M1 macrophages and decreases Treg cells in the tumor microenvironment. These changes in tumor microenvironment is consistent with reduced tumor growth by combination therapy. The most intriguing part of the study is the determination of effect of Nifuroxazide on PD-L1 expression in the context of radiotherapy. Considering Nifuroxazide is a stat3 activation inhibitor and stat3 inhibition leads to reduced expression of PD-L1, one would expect Nifuroxazide decreases PD-L1 expression through stat3. However, they find the effect of Nifuroxazide on PD-L1 is dependent on GSK3 mediated Proteasome pathways and independent of stat3, in the given experimental context. To determine the relevance to human hepatocellular carcinoma, they also measured the PD-L1 expression in human tumor tissues of HCC patients pre- and post-radiotherapy. The increased PD-L1 expression level in HCC after radiotherapy is impressive.

Overall, the data are convincing and supportive to the conclusions.

Strengths:

1. Novel finding: Identified novel mechanism underlying effect of Nifuroxazide on PD-L1 expression in hepatocellular carcinoma cells in the context of radiotherapy.

2. Comprehensive experimental approaches: using different approaches to prove same finding. For example, Fig4, both IHC and WB were used. Fig5. Both IF and WB were used.

3. Human disease relevance: Compared observations in mice with human tumor samples.

---

## [Referee Report · Reviewer #3 (Public Review)]

Summary:

In this study, the authors investigated the potential of nifuroxazide to enhance responsiveness to radiotherapy, employing both an in vitro cell culture system and an in vivo syngeneic mouse tumor model.

Strengths:

The researchers conducted a series of experiments to elucidate the role of nifuroxazide in facilitating the radiotherapy-induced reduction of proliferation, migration, and invasion of HepG2 cells.

Weaknesses:

The evidence supporting the claim that nifuroxazide contributes to the degradation of radiotherapy-induced upregulation of PD-L1 via the ubiquitin-proteasome pathway is still relatively weak.

---

## [Author Response]

The following is the authors’ response to the original reviews.

**Reviewer #1 (Public Review):**
The authors found that nifuroxazide has the potential to augment the efficacy of radiotherapy in HCC by reducing PD-L1 expression. This effect may be attributed to increased degradation of PD-L1 through the ubiquitination-proteasome pathway. The paper provides new ideas and insights to improve treatment effectiveness, however, there are additional points that could be addressed.The paper highlights that the combination of nifuroxazide increases tumor cell apoptosis. A discussion regarding the potential crosstalk or regulatory mechanisms between apoptotic pathways and PD-L1 expression would be valuable.

Response: Thank you very much for your suggestion. Research has shown that regulating the STAT3/PD-L1 pathway can effectively increase apoptosis in lung cancer cells (1). Our study confirmed that nifuroxazide can effectively inhibit the expression of p-STAT3 and PD-L1 in liver cancer cells, which may be the reason for the increased apoptosis of these cells. We have added relevant descriptions in the discussion.

The benefits and advantages of nifuroxazide combination could be compared to the current clinical treatment options.

Response: Thank you greatly for your insightful feedback. The primary objective of this study is to explore whether nifuroxazide can effectively enhance the degradation of PD-L1, thereby increasing the radiosensitivity of HCC. Our research reveals that compared to radiation therapy alone, combination therapy involving nifuroxazide and radiation significantly inhibits tumor growth in mice and boosts the anti-tumor immune response. This finding could potentially provide a valuable strategy for patients who exhibit resistance to radiation therapy in clinical practice. Moreover, clinical trial investigations have demonstrated that nivolumab, a PD-1 monoclonal antibody, when combined with radiation therapy for HCC, exhibits promising safety and efficacy (2). This evidence supports the future application of nifuroxazide in the treatment of HCC.However, to reach this objective, we must continue to conduct extensive research, including comparing nifuroxazide with existing therapies in clinical practice. We believe that nifuroxazide not only significantly inhibits the expression of PD-L1 protein in HCC cells but also functions as a PD-L1 inhibitor. Furthermore, it effectively curbs the proliferation and migration of HCC cells, induces tumor cell apoptosis, and may exhibit enhanced anti-tumor effects, making it a promising candidate for clinical use. We have incorporated relevant discussion content in the article to address these points.

**Reviewer #2 (Public Review):**
Summary:Zhao et al. aimed to explore an important question - how to overcome the resistance of hepatocellular carcinoma cells to radiotherapy? Given that the immune-suppressive microenvironment is a major mechanism underlying resistance to radiotherapy, they reasoned that a drug that blocks the PD-1/PD-L1 pathway could improve the efficacy of radiation therapy and chose to investigate the effect of Nifuroxazide, an inhibitor of stat3 activation, on radiotherapy efficacy in treating hepatocellular carcinoma cells. From in vitro experiments, they find combination treatment (Nifuroxazide+ radiotherapy) increases apoptosis and reduces proliferation and migration, in comparison to radiotherapy alone. From in vivo experiments, they demonstrate that combined treatment reduces the size and weight of tumors in vivo and enhances mice survival. These data indicate a better efficacy of combination therapy compared to radiotherapy alone. Moreover, they also determined the effect of combination therapy on tumor microenvironment as well as peripheral immune response. They find that combination therapy increases infiltration of CD4+ and CD8+ cells as well as M1 macrophages in the tumor microenvironment. Interestingly, they find that the ratio of Treg cells in spleen is increased by radiotherapy but decreased by Nifuroxazide. Considering the immune-suppressive role of Treg cells, this finding is consistent with reduced tumor growth by combination therapy. However, it is unclear whether the combined therapy affects the ratio of Treg cells in the tumors or not. The most intriguing part of the study is the determination of the effect of Nifuroxazide on PD-L1 expression in the context of radiotherapy. Considering Nifuroxazide is a stat3 activation inhibitor and stat3 inhibition leads to reduced expression of PD-L1, one would expect Nifuroxazide decreases PD-L1 expression through stat3. However, they found that the effect of Nifuroxazide on PD-L1 is dependent on GSK3 mediated Proteasome pathways and independent of stat3, in the given experimental context. To determine the relevance to human hepatocellular carcinoma, they also measured the PD-L1 expression in human tumor tissues of HCC patients pre- and post-radiotherapy. The increased PD-L1 expression level in HCC after radiotherapy is impressive. However, it is unclear whether the patients being selected in the study had resistant disease to radiotherapy or not.Overall, the data are convincing and supportive to the conclusions.Strengths:1. Novel finding: Identified novel mechanism underlying the effect of Nifuroxazide on PD-L1 expression in hepatocellular carcinoma cells in the context of radiotherapy.1. Comprehensive experimental approaches: using different approaches to prove the same finding. For example, in Fig 4, both IHC and WB were used. In Fig 5, both IF and WB were used.1. Human disease relevance: Compared observations in mice with human tumor samples.The question in the summary, “However, it is unclear whether the combined therapy affects the ratio of Treg cells in the tumors or not”.

Response: Thank you very much for your valuable feedback. We have included additional flow cytometry results regarding the expression of relevant Treg cells (CD4+CD25+Foxp3+ T lymphocytes) in tumor tissues (Supplementary Fig 2). Our findings indicate that the number of Treg cells in tumor tissues significantly decreased following combination therapy with nifuroxazide and radiotherapy.

The question in the summary, “However, it is unclear whether the patients being selected in the study had resistant disease to radiotherapy or not”.

Response: Thank you very much for your valuable feedback. All the HCC patients selected in this study experienced recurrence after radiation treatment.

Weaknesses:1. It is hard to tell whether the observed phenotype and mechanism are generic or specific to the limited cell lines used in the study. The in vitro experiments were performed in one human cell line and the in vivo experiments were performed in one mouse cell line.

Response: Thank you very much for your feedback. We have included additional experimental data from another human cell line Huh7 (Supplementary Fig 3).

1. The study did not distinguish the effect of increased radiosensitivity by nifuroxazide from combined anti-tumor effects by two different treatments.

Response: Thank you greatly for your insightful feedback. In this study, we primarily compared the antitumor effects of nifuroxazide combined with radiotherapy versus either nifuroxazide or radiotherapy alone, and confirmed that the combined treatment demonstrated a more potent anti-hepatocellular carcinoma effect compared to single therapy. Furthermore, to achieve the goal of utilizing nifuroxazide for the treatment of clinical hepatocellular carcinoma, additional research is necessary, including comparisons with other clinically established therapies. We have also incorporated relevant discussions in our analysis.

**Reviewer #3 (Public Review):**
Summary:In this study, the authors embarked on an exploration of how nifuroxazide could enhance the responsiveness to radiotherapy by employing both an in vitro cell culture system and an in vivo mouse tumor model.Strengths:The researchers conducted an array of experiments aimed at revealing the function of nifuroxazide in aiding the radiotherapy-induced reduction of proliferation, migration, and invasion of HepG2 cells.Weaknesses:The authors did not provide the molecular mechanism through which nifuroxazide collaborates with radiotherapy to effectively curtail the proliferation, migration, and invasion of HCC cells. Moreover, the evidence supporting the assertion that nifuroxazide contributes to the degradation of radiotherapy-induced upregulation of PD-L1 via the ubiquitin-proteasome pathway appears to be insufficient. Importantly, further validation of this discovery should involve the utilization of an additional syngeneic mouse HCC tumor model or an orthotopic HCC tumor model.

Response: Thank you very much for your insightful comments. Nifuroxazide has been demonstrated to inhibit the expression of p-STAT3, thereby suppressing tumor cell proliferation and migration (3, 4). In our study, we observed that after 48 hours of treatment with Nifuroxazide, the expression of p-STAT3 in irradiated cells was significantly inhibited. Furthermore, compared to radiation alone, combined Nifuroxazide and radiotherapy resulted in a more pronounced decrease in PCNA expression. Simultaneously, we performed additional detection of migration-related protein MMP2 expression (revised Fig 2B), confirming that combined Nifuroxazide and radiotherapy led to a more significant inhibition of MMP2 expression. These findings suggest that the combined treatment may be responsible for the synergistic suppression of HCC cell proliferation and migration. We have included relevant discussions in our manuscript.

Our initial results indicate that Nifuroxazide inhibits the expression of PD-L1 at the protein level, but does not affect its mRNA level. Interestingly, upon treatment with a proteasome inhibitor MG132, the inhibitory effect of Nifuroxazide on PD-L1 was eliminated, suggesting that Nifuroxazide may enhance the degradation of PD-L1 protein. Our experiments have demonstrated the inhibitory effect of Nifuroxazide on PD-L1 in both human and mouse cell lines. However, to translate these findings into clinical application for the treatment of hepatocellular carcinoma, additional research is necessary, including validation in genetically engineered mouse models of HCC. We have addressed these points in the discussion section of our manuscript.

**Reviewer #1 (Recommendations For The Authors):**
1. Please improve the quality of Figure 3E. It is hard to figure out the bar and details.

Response: Thank you for your valuable feedback. We have meticulously revised the figures to enhance their clarity and presentation (revised Fig 3E).

1. In Figure 7E, please elucidate the methods used for calculating the amount of PD-L1 mRNA level. Please adjust the picture angle and label the marker size on the left as well

Response: Thank you for your feedback. We have incorporated a method for calculating PD-L1 mRNA levels and revised the corresponding figures accordingly (revised Fig 7E).

**Reviewer #2 (Recommendations For The Authors):**
Questions:1. What is the advantage of using a combination of nifuroxazide and radiotherapy in comparison to using a combination of anti-PD1/PDL1 and radiotherapy?

Response: Thank you very much for your insightful comments. We believe that the advantage of nifuroxazide over PD-1 or PD-L1 antibodies lies in its ability not only to effectively inhibit PD-L1 expression but also to suppress tumor cell proliferation, migration, and promote cell apoptosis (Supplementary Fig 1). We have also expanded on these aspects in the discussion section of the manuscript.

1. For the characterization of tumor microenvironment and immune cells in the spleen, were the same cell populations being investigated? What about NK and Treg cells in tumors? What about M1 macrophages in spleen?

Response: Thank you very much for your insightful suggestion. We have measured the infiltration of NK and Treg cells in tumor tissues (Supplementary Fig 2), as well as the abundance of M1 macrophages (revised Fig 6) in the spleen, and provided additional relevant data to strengthen our study.

Other comments:1. The data in Fig 1 is solid. However, it is hard to distinguish the effect of increased radiosensitivity by nifuroxazide from combined anti-tumor effects by two different treatments. The anti-tumor role of Nifuroxazide has been reported in melanoma, colorectal carcinoma, and hepatocellular carcinoma previously (PMID: 26830149; 28055016, 26154152). Therefore, the increased apoptosis and decreased proliferation and migration could be caused by nifuroxazide and not related to the sensitivity of cells to radiation therapy.

Response: Thank you very much for your constructive feedback. As you suggested, the anti-tumor role of nifuroxazide has been reported. However, the innovation of our study does not lie in confirming its antitumor effects but rather in demonstrating how nifuroxazide can enhance radiotherapy's efficacy in treating hepatocellular carcinoma by inhibiting PD-L1 levels.

We compared the efficacy of combined therapy versus radiotherapy and found that compared to radiation alone, combined therapy more significantly inhibited hepatocellular carcinoma cell proliferation and migration. In our animal model, we compared the therapeutic effects of combined therapy, nifuroxazide, and radiotherapy on hepatocellular carcinoma-bearing mice. We observed that compared to individual treatment groups, combined therapy more profoundly suppressed tumor growth and enhanced the antitumor effects in the mice.

In response to your feedback, we have expanded the discussion on the impact of combined therapy versus nifuroxazide or radiotherapy on hepatocellular carcinoma cell proliferation, migration, and apoptosis (Supplementary Fig 1). The data show that compared to either individual therapy, combined therapy further inhibited cell proliferation and migration while promoting apoptosis.

1. There is no direct evidence to show the improved efficacy of radiation therapy by nifuroxazide through the degradation of PD-L1.

Response: Thank you very much for your valuable suggestions. In our cell experiments, we found that nifuroxazide inhibits the increased expression of PD-L1 in cells induced by radiation therapy, and this inhibitory effect is counteracted when using the proteasome inhibitor MG132. Therefore, we speculate that nifuroxazide may inhibit PD-L1 expression through a proteasome-dependent mechanism. To better reflect this, we have revised the title of our manuscript to "Nifuroxazide Suppresses PD-L1 Expression and Enhances the Efficacy of Radiotherapy in Hepatocellular Carcinoma."

1. "The oncogene Stat3.....was effectively inhibited by radiotherapy in cells" - this sentence may be rephrased to make the point clear. The authors might mean to say "activation of the oncogene stat3....""The results demonstrated that the combination therapy increased the expression of PARP," the authors might mean to say "expression of c-PARP"

Response: Thank you very much for your feedback. We have revised the relevant sentence descriptions to improve clarity and accuracy.

1. "histomorphology significantly improved after the treatment with nifuroxazide and radiation therapy (Fig 3E)." How to define "improved histomorphology"? The authors may want to provide more details to clarify "improved".

Response: Thank you very much for your feedback. We have revised the relevant sentence descriptions to improve clarity and accuracy.

1. In addition to normalizing protein expression by tubulin, the authors may consider normalizing p-stat3 expression level by stat3.

Response: Thank you very much for your feedback. We have conducted a quantitative analysis of the expression levels of p-STAT3 and STAT3 (revised Fig 2A).

1. Figure 3C and D, using a different color to represent each group might help the readers to better differentiate each group.

Response: Thank you very much for your feedback. Following your suggestion, we have revised the figures accordingly (revised Fig 3C and 3D).

**Reviewer #3 (Recommendations For The Authors):**
In this study, the authors revealed the pivotal role of nifuroxazide in augmenting the efficacy of radiotherapy. This was evidenced by its synergistic effect in suppressing the proliferation and migratory capabilities of HCC cells, alongside its capacity to induce apoptosis in these cells. Furthermore, their findings underscored the substantial synergy between nifuroxazide and radiotherapy in retarding tumor growth, thereby extending survival rates in a tumor-bearing murine model. Moreover, the authors observed that nifuroxazide combined with radiotherapy significantly increases the tumor-infiltrating CD4+ T cells, CD8+ T cells, and M1 macrophages. Finally, the authors found that nifuroxazide countered the radiotherapy-induced upregulation of PD-L1 through the ubiquitin-proteasome pathway. However, the evidence for supporting the main claims is only partially supported. The following are my concerns and suggestions.1. In Figures 1 and 2, the authors convincingly demonstrate the synergistic impact of nifuroxazide and radiotherapy on curtailing the proliferation, colony formation, and migratory capabilities of HCC cells, while also instigating apoptosis in these cells. However, the underlying molecular mechanism remains elusive. A recent study highlighted nifuroxazide's potential to impede the proliferation of glioblastoma cells and induce apoptosis via the MAP3K1/JAK2/STAT3 pathway (Wang X., et al., Int Immunopharmacol. 2023 May;118:109987. doi: 10.1016/j.intimp.2023.109987). It would be valuable for the authors to investigate whether nifuroxazide employs a similar molecular mechanism to regulate proliferation and apoptosis in the context of HCC. This could offer deeper insights into the mechanisms at play in their observed effects.

Response: Thank you very much for your insightful comments. As you pointed out, previous studies have reported that nifuroxazide exerts antitumor effects by inhibiting the STAT3 pathway. However, in our experiments, we observed that radiation therapy significantly increased the expression of PD-L1, but showed a trend of decreased p-STAT3 expression. Therefore, we believe that nifuroxazide does not inhibit PD-L1 expression through the STAT3 pathway. Subsequently, our further research revealed that the inhibitory effect of nifuroxazide on PD-L1 can be counteracted by a proteasome inhibitor. Thus, we propose that nifuroxazide inhibits PD-L1 expression through a proteasome-dependent mechanism, thereby enhancing the efficacy of radiation therapy in hepatocellular carcinoma.

1. Figures 1 and 2 solely rely on the HepG2 cell line to establish their conclusions. To validate these findings robustly, it is recommended that another HCC cell line be included in the study. This additional cell line will contribute to the generalizability and reliability of the results, enhancing the overall credibility of the study's conclusions.

Response: Thank you very much for your suggestion. We have included additional experimental results with the relevant cell line Huh7 (supplementary Fig 3).

1. Figure 3 demonstrates the use of only one syngeneic mouse H22 tumor model. To ensure the robustness and validity of this finding, it would be advisable to incorporate at least one more syngeneic mouse HCC tumor model or even an orthotopic mouse tumor model. The inclusion of additional models would bolster the significance and reliability of the observed results, contributing to a more comprehensive understanding of the phenomenon under investigation.

Response: Thank you for your valuable suggestion. In the H22 mouse tumor model, we conducted relevant assessments of survival rate and tumor growth. The results confirm that the combination of nifuroxazide and radiation therapy exhibits a promising synergistic antitumor effect. However, to achieve the goal of applying nifuroxazide combined with radiation therapy for the treatment of clinical hepatocellular carcinoma, we still need to undertake extensive research, including validation on genetically identical mouse HCC tumor models. We have also included relevant discussions in our ongoing discussions.

1. In Figure 5, employing an alternative method, such as the flow cytometry assay, to analyze and corroborate the tumor-infiltrating immune cell profiling following various treatments would enhance the rigor of the study. This additional approach would provide a complementary perspective and validate the findings, strengthening the overall reliability and impact of the results presented.

Response: Thank you for your insightful suggestion. We have included additional experimental data to strengthen our study (supplementary Fig 2).

1. In Figure 7, the conclusion drawn regarding nifuroxazide's impact on PD-L1 expression through ubiquitination-proteasome mechanisms seems to lack the robust evidence needed to firmly establish nifuroxazide's role in regulating PD-L1 ubiquitination. To reinforce this aspect of the study, the authors may conduct comprehensive in vitro and in vivo ubiquitination assays. Performing these assays would offer direct insights into whether nifuroxazide genuinely influences PD-L1 ubiquitination, thus fortifying the credibility and importance of the reported findings.

Response: Thank you for your valuable feedback. Our initial findings suggest that nifuroxazide inhibits the expression of PD-L1 protein levels, but does not affect the mRNA levels. Moreover, upon treatment with the proteasome inhibitor MG132, the inhibitory effect of nifuroxazide on PD-L1 was found to be abolished. Concurrently, we observed that nifuroxazide significantly enhances GSK-3β expression in both cell and animal experiments. Consequently, we propose that nifuroxazide augments the degradation of PD-L1 protein.

1. Statistical methods should be included in the captions of all the figures with statistical graphs. The size of the scale should be supplemented with a description in the captions.

Response: Thank you for your valuable suggestion. We have made the appropriate modifications to our study based on your recommendations.

1. Considering the outcomes presented in the study, it appears that the title "Nifuroxazide enhances radiotherapy efficacy against hepatocellular carcinoma by upregulating PD-L1 degradation via the ubiquitin-proteasome pathway" may not accurately reflect the findings.

Response: Thank you for your insightful feedback. We have revised the title to read, "Inhibitory Effects of Nifuroxazide on PD-L1 Expression and Enhanced Radiotherapy Efficacy in Hepatocellular Carcinoma".

References

1. Xie C, Zhou X, Liang C, Li X, Ge M, Chen Y, et al. Apatinib triggers autophagic and apoptotic cell death via VEGFR2/STAT3/PD-L1 and ROS/Nrf2/p62 signaling in lung cancer. Journal of experimental & clinical cancer research : CR. 2021;40(1):266. doi: 10.1186/s13046-021-02069-4.

2. de la Torre-Alaez M, Matilla A, Varela M, Inarrairaegui M, Reig M, Lledo JL, et al. Nivolumab after selective internal radiation therapy for the treatment of hepatocellular carcinoma: a phase 2, single-arm study. Journal for immunotherapy of cancer. 2022;10(11). doi: 10.1136/jitc-2022-005457.

3. Yang F, Hu M, Lei Q, Xia Y, Zhu Y, Song X, et al. Nifuroxazide induces apoptosis and impairs pulmonary metastasis in breast cancer model. Cell Death Dis. 2015;6(3):e1701. doi: 10.1038/cddis.2015.63.

4. Nelson EA, Walker SR, Kepich A, Gashin LB, Hideshima T, Ikeda H, et al. Nifuroxazide inhibits survival of multiple myeloma cells by directly inhibiting STAT3. Blood. 2008;112(13):5095-102. doi: 10.1182/blood-2007-12-129718.